# Phonetic differences between affirmative and feedback head nods in German Sign Language (DGS): A pose estimation study

**Anastasia Bauer**[1]*, **Anna Kuder**[1], **Marc Schulder**[2], **Job Schepens**[1]

**1** Department of Linguistics, General Linguistics, University of Cologne, Cologne, Germany, **2** Institute for German Sign Language and Communication of the Deaf, University of Hamburg, Hamburg, Germany

* anastasia.bauer@uni-koeln.de

## Abstract

This study investigates head nods in natural dyadic German Sign Language (DGS) interaction, with the aim of finding whether head nods serving different functions vary in their phonetic characteristics. Earlier research on spoken and sign language interaction has revealed that head nods vary in the form of the movement. However, most claims about the phonetic properties of head nods have been based on manual annotation without reference to naturalistic text types and the head nods produced by the addressee have been largely ignored. There is a lack of detailed information about the phonetic properties of the addressee's head nods and their interaction with manual cues in DGS as well as in other sign languages, and the existence of a form-function relationship of head nods remains uncertain. We hypothesize that head nods functioning in the context of affirmation differ from those signaling feedback in their form and the co-occurrence with manual items. To test the hypothesis, we apply OpenPose, a computer vision toolkit, to extract head nod measurements from video recordings and examine head nods in terms of their duration, amplitude and velocity. We describe the basic phonetic properties of head nods in DGS and their interaction with manual items in naturalistic corpus data. Our results show that phonetic properties of affirmative nods differ from those of feedback nods. Feedback nods appear to be on average slower in production and smaller in amplitude than affirmation nods, and they are commonly produced without a co-occurring manual element. We attribute the variations in phonetic properties to the distinct roles these cues fulfill in turn-taking system. This research underlines the importance of non-manual cues in shaping the turn-taking system of sign languages, establishing the links between such research fields as sign language linguistics, conversational analysis, quantitative linguistics and computer vision.

## Introduction

Head nod is one of the most commonly produced bodily signals in interaction. It can be defined as a vertical oscillating movement of the head, usually going up and down in a smooth manner without pausing. Both signers/speakers as well as addressees produce head nods

**Data Availability Statement:** The following sentence is included here and in the manuscript, section "Supporting information": The annotations and code produced in the context of this article are

publicly available at https://doi.org/10.5281/zenodo.10838847. The archive contains the following: • the manual annotations created for the article in ELAN (.eaf) format; • the Python code used for quantitative analysis of the annotations with the aid of OpenPose data from MY DGS – annotated; • the R code used to perform the statistical analyses discussed in the article as well as the depicted graphs.

**Funding:** German Science Foundation (DFG) SPP 2392 Visual Communication (ViCom).

**Competing interests:** The authors have declared that no competing interests exist.

during everyday face-to-face communication. Head nod is associated with a number of different functions in communicative interaction [1–6]. While head nodding is extremely prevalent in natural conversations, there is limited information available regarding its phonetic characteristics and its interaction with other (non-)manual elements in signed interaction, especially with regard to the head nods produced by the addressee.

This is partly due to methodological challenges, since the majority of previous qualitative analyses of sign languages rely on manual annotation of non-manuals which is known to be a very time consuming and expensive task with a tendency towards low inter-annotator agreement [7]. Recent technological advances in computer vision (CV) and machine learning have resulted in applications that promise to provide an important step towards facilitating this process. There is a number of tools that allow automatic detection and tracking of human body movements in video recordings. Especially since 2020, more and more studies make use of these techniques in gesture and sign language research [8–12].

Despite this progress, there is only a limited number of studies which have applied CV to study non-manual activity in sign languages, especially to analyze the phonetic properties of linguistically relevant non-manual movements. Kimmelman and colleagues have applied OpenPose [13] and OpenFace [14] for the analysis of two non-manual signals: eyebrow movements in Kazakh-Russian Sign Language [15, 16] and headshakes expressing negation in Russian Sign Language [17]. They show that CV tools are suitable for linguistic analysis of non-manual cues in sign languages when combined with prior manual annotations of the data. We build on their results and use the combination of two methods in this study: manual annotation of head nods in the data and the extraction of measurements with CV. As research shows, CV tools are suitable to extract phonetic measurements of head movements, but they are not yet able to detect head movements in the data automatically [17]. Automatic detection of head movements in video-recorded face-to-face dyadic conversations seems, at the current state, not to be accurate and reliable enough. In a study by Paggio and colleagues [18] head movement recognition software trained on spoken language conversational video data annotated frame-wise with visual and acoustic features was found to predict head movement only with 0.75 accuracy. A number of especially subtle head movements, as is the case with some forms of head nods, are likely not to be detected in the data. We, therefore, follow Kimmelman and colleagues in combining the measurements extracted with OpenPose with manual annotations of the same video recordings to provide an analysis of phonetic properties of head nod, one of the most prevalent non-manual cues present in naturalistic conversational data. Recordings are drawn from the Public DGS Corpus [19].

## Previous research on head nods

Head nods are vertical movements of the head. A prototypical movement path of a head nod is an upward movement, followed by the downward movement and/or a subsequent movement toward a neutral position. These three movements represent a single head nod. Head nods vary in their form and may come as single nods or in a repeated fashion. They are usually produced in a smooth manner without pausing. Both signers/speakers as well as addressees produce a great number of head nods during everyday face-to-face interactions.

Previous research on head nods in spoken and signed languages suggests that head nods may perform a multitude of diverse functions in interaction such as affirmation, acceptance, agreement, emphasis, affiliation, existence, assertion and feedback among many others [1–5, 20–25]. The manifestations of some of these functions exhibit variability across diverse cultural contexts (e.g. agreement or affirmation is associated with a lateral head movement in the Bulgarian cultural tradition [26]).

The functions vary primarily according to whether the head nods are produced by one who is speaking or signing or the one who is being addressed [27]. While head nods with the emphatic function are unlikely to be produced by the addressees, the feedback head nods are improbable to be produced by the person holding the floor. The most common function carried out by head nods in interaction seems to be that of feedback [4, 28].

The phonetic features of head nods (such as the amplitude, velocity, direction or the number of repetitions) have been described as a basis for differentiation between various functions of polysemous head movements (e.g. see [27] for head nods in spoken political debates). However, the manual determination of such phonetic features is laborious and, therefore, challenging to provide at scales suitable for quantitative analysis.

While most of the research on head nods in spoken languages has concentrated on the use of this movement produced by the addressee [2, 4, 24, 28–30], previous sign language research has rather focused on the head nods produced by the signer and paid little attention to the addressee's nodding. There is only a small number of observations of addressee's head nod in face-to-face signed interaction [6, 31, 32] and we lack detailed information about the phonetic properties of head nods and their interaction with other (non-)manual cues in signed conversations.

Non-manual elements have received great deal of attention from the sign language linguists and prompted extensive research in the last decades. Linguistically significant non-manuals have been assigned various morphological, syntactic and prosodic functions [22, 23, 33, 34]. Whereas headshake generated significant interest among sign linguists [35–39], head nod has received far less attention in the sign language literature (but see [6, 21, 40, 41]). Head nods have been primarily investigated while being accompanied by manual signs produced by the signer [23]. Researchers working on sign languages have so far described head nods as aspectual or prosodic non-manual markers which signal clause and constituent boundaries, mark phonological and intonational phrases in signed narratives or express the truth value of a proposition (adding assertive or existential character) [21, 40–44].

Sign language linguists have noticed that head nods may vary in their form (i.e. single vs. repeated nods) and also in their phonetic properties, such as amplitude, velocity or the direction of the movement [21, 40, 41]. Describing head nods as used by the person signing, Liddell observes that fast, slow and deep head nods which co-occur with manual signs may mark assertion, emphasis or existence [21]. Wilbur suggests that single head nods are used rather as boundary markers, whereas slow and deliberate head nods are focus markers, co-occurring with lexical signs [22]. Repetitive head nods have been reported to convey the signer's commitment to the truth of the assertion [22]. Distinguishing between confirmative and assertive head nods in ÖGS (Austrian Sign Language), Lackner [41] notes that nods may have various size and speed of the movement as well as variation with regard to their co-articulation with lexical signs. She notes that confirmative nods are small and can be produced without any lexical signs. Assertive nodding is described as fast, small movement that co-occurs with a syntactic constituent forming a single lexical element, as, for instance, with assertive modal verbs like CAN, MUST, or SHOULD, which are usually accompanied by one or more head nods. In her study, Lackner [41] finds that form of the head nod is subject to variation (but both fast, small nods and slow, large nods may occur with the same function). Nods with regular movement are usually associated with giving assertive answers, assertions, confirmations, marking positive contrast. Nods produced with slow movement are markers of assertion, confirmation, and epistemic modality. The last function may also be conveyed, according to Lackner, by nods produced with fast, small movements. Lastly, nods produced 'in a trembling way' convey the timitive epistemic modality [41]. Although Lackner attempts to, she does not provide a clear-cut pairing of different types of nods with their various functions. All of the studies described

above focus on head nods produced by the active signer and they rely on laborious manual annotation of head movements.

Puupponen and colleagues initiated a study to analyze the phonetic properties of head movements in FinSL by using the most accurate type of data available to them for the phonetic analysis of sign language [6]. They analyze the kinematic properties of signers' movements in data recorded with motion capture (mocap) technology. While being a method with high reliability, mocap is a very time-consuming and expensive technique. As a result, the authors conduct their analyses on the basis of only two FinSL dialogues consisting of 2:15 minutes of mocap data with two native FinSL signers. Within this dataset, head nod is the most common type of head movement. The study differentiates between two forms of nods: single (nods) and continuous (nodding) head movements and the six different functions: emphasis, boundary marking, affirmation, interrogative, copying and indicating. Comparing the measurements from the 70 nods (36 single and 34 continuous nods), Puupponen and colleagues conduct the analysis of phonetic properties of these head movements and report the average duration of a single nod (1.3 seconds) and of a continuous nod (2.1 seconds) and the average amplitude of movement in the dimension of depth (43 millimeters). Without reporting the variation in duration, they find a considerable variation in the movement amplitude of different nods, which varies from 15 to 55 millimeters. It is the first study to pay attention to the phonetic properties of all nods in signed interaction, including the nods produced by the addressee using a more reliable technology than manual annotation. The function referred to as affirmation in that study appears to be the most frequent one in the inspected data. This is not surprising, since the authors do not distinguish between affirmative and feedback head nods and group them into a single category (called 'A-function').

## The current study

In this study we analyze head nods in German Sign Language (DGS, *Deutsche Gebärdensprache*), using the Public DGS Corpus [19]. We manually identify head nods in the video-recorded DGS data and analyze them in terms of their co-articulation with (non-)lexical manual forms and their phonetic properties based on the measurements extracted from OpenPose. The aim of the study is two-fold: (1) to provide a description of the basic phonetic properties of a head nod in DGS and (2) to examine whether head nods serving different functions vary in their phonetic characteristics.

We first manually identified and annotated head nods in the data and then assigned them to various functions. Contrary to the previous sign language research, which has primarily focused on head nods produced by the signer in narration data, we center our attention on addressee's head nods in conversational data. While head nods serve various functions [6, 21, 27, 45], this study specifically concentrates on only two of them: affirmation and feedback. We investigate whether head nods fulfilling these two functions in discourse differ significantly in their phonetic properties. We assume that these types of nods would be different in a number of features.

We use the term 'affirmation' to describe somebody's positive reaction to a question. If a person produces a head nod while confirming, approving or agreeing with the preceding question in the discourse, we categorize it as an affirmative nod (see the example in section *Annotation*).

We define a nod as fulfilling the function of feedback, when it functions as an interactional behavior that displays interlocutors' perception or understanding of the course of the conversation. We focus on the very common feedback mechanisms, which can signal a non-uptake of a conversation turn, acknowledge a prior statement or demonstrate understanding of the

information represented by another signer, signal a piece of information as new or even evaluate it. Some of these feedback signals are referred to as continuers, backchannels, minimal responses, reactive or response tokens in the literature [31, 32, 46–51]. We do not differentiate between various subcategories of feedback, such as continuers, acknowledgements, newsmarkers or assessments (see [49]), but rather use the term 'feedback' to broadly refer to manual or non-manual cues produced by the addressee, who reacts to the proposition produced by the other signer and thus signals participation and engagement in the conversation (see section *Annotation* for an example).

There are reasons to assume that affirmative nods would differ from feedback nods in their kinematic characteristics and use. Research on feedback signals in sign languages suggests that manual signs are used extremely rarely for signaling feedback in signed conversations [31, 52, 53]. Feedback mechanisms in sign languages appear to be predominantly non-manual. It can be inferred that feedback nods would typically occur without any co-occurring manual signs. A study by Hadar and colleagues [28] on head movements in spoken English conversation suggests that the listeners' head nods are more subtle. Based on their analysis of 15 minutes of spoken English conversation, the authors claim that head movements anticipating a turn have a higher mean amplitude than the head movements signaling a non-uptake of a conversation turn. Nods with an affirmative meaning in Polish task-oriented dialogues have been observed to contain a high amplitude [54]. Therefore, it is reasonable to assume that head nods signaling feedback in DGS will be different in their phonetic properties from the affirmation nods, which typically initiate a turn by providing a response to a question. Although work on conversation analysis in DGS and other sign languages has been scarce and there is little research on signed turn-taking behavior and the use of head nods, it has been mentioned in the seminal work by Baker [55] and in later study by Napier and colleagues [56] that a commonly employed strategy for initiating a turn involves increasing the size and quantity of head nodding.

Therefore, drawing on insights from the literature and our experience with annotation, we formulate the following hypotheses:

1. Head nods functioning as affirmative responses will be larger in amplitude, shorter in duration and faster in their production than head nods which signal feedback in interaction. Feedback nods are expected to have a smaller amplitude but longer duration of the movement and to be produced more slowly.

2. Affirmation nods will co-occur with manual items more often than is the case for feedback nods.

We assume that feedback nods are usually produced without co-occurring manual lexical or non-lexical components in order to not interrupt the signer's turn following principles of minimal effort [57]. Affirmation nods are anticipated to be more visible (to be larger in amplitude and faster and to co-occur with manual signs in signed interaction) in order to indicate the addressee's readiness to take over the conversational turn and answer the question stated. In a study on the correlation between head nods and vocalizations in spoken English, Dittmann and Llewellyn observed that addressees tend to combine a head nod with vocalization when responding to questions, rather than in feedback contexts [58]. This combination of modalities may serve as a cue to the interlocutor that the addressee now wishes to contribute to the conversation.

The rest of this paper is organized as follows: Section *Materials and methods* provides an overview of the data and the methods used for the current study. Section *Results* describes the phonetic properties of a mean head nod in DGS. Moreover, it features a report on phonetic

features which differ between head nods functioning as feedback signals or as affirmation markers. In section *Discussion & outlook* we propose that the variation in phonetic properties is important for shaping the rapid turn-taking system and ensuring successful communication. In the same section we also point out the limitations of this study and possible directions for future research.

## Materials and methods

To investigate the phonetic characteristics of head nods in DGS, we implemented the following steps, which we elaborate on in the upcoming sections:

- manual annotation of all head nods in a subset of the Public DGS Corpus [19] with the use of ELAN software [59];

- assigning the functions of 'affirmation', 'feedback' or 'other' to the annotated head nods;

- extracting the head nod measurements using OpenPose [13] and

- quantitative analysis of a subset of the measurements using R and RStudio [60].

  All annotations and analysis code are publicly available as part of the S1 File.

### Annotation

To analyze the phonetic properties of head nods in DGS, we conducted the following three study-specific annotations: Form, Function and Turn-taking behavior. Prioritizing the research practices of openness, transparency, replicability and quality control, we make all annotations and analysis code produced for this article publicly available (see section S1 File) to allow other researchers to reproduce the results.

**Form.**   Head nods were annotated by the first and second author. By head nods we mean all oscillatory up and down movements of the head produced in the vertical axis, while the motion in the horizontal axis is limited. A nod is produced when the head leaves the neutral position to go either up or down, then moves in the opposite direction and crosses the neutral position to then go back in the same direction it initially moved in in order to go back to its rest position at the end of the movement. In this study, we do not differentiate nods based on the direction of movement, but for future research, the OpenPose data could be used to infer this information. Each head nod thus consists of (at least one) pair of a peak and a trough, regardless of the order these two appear in. If the up movement was held for a while followed by the movement towards a neutral position, it was labeled as head back tilt and was not considered a head nod in the current analysis (similar treatment of this head movement can be also found in other authors' work, e.g. [20, p. 47] and [41, p. 12]).

The head nods identified in the DGS data were annotated for form according to the following practices:

- the onset of each tag was counted as the first frame in which the head movement out of the neutral position (either up or down) was noticeable with the bare eye;

- the offset of each tag was counted as the first frame in which the head returned to the neutral position;

- two tags were identified as separate movements if the offset of one tag and the onset of the next were at least 300 ms apart (this number was chosen as [61, 62], found it to be the approximate minimum length of time that naïve observers need to consistently identify a

cessation of movement). If the 300 ms threshold was not met the identified movement was annotated as continuous nodding.

The tags used in the annotation process of the form of the head nod are summed up in Table 1 below.

To assess the reliability of the annotation process, calculation of inter-annotator reliability (IAR) was conducted by comparing an annotation effort of a single file by two independent annotators. Both annotators fully and independently of each other annotated the form of the nodding activities produced by one signer in transcript ber_07_deaf_event to see whether the annotation scheme was consistently applied by both of them. In this file the first annotator identified 56 occurrences of nodding and the second annotator identified 46 occurrences of nodding (produced by the inspected informant). As annotations were done from scratch and did not rely on any annotations previously provided within the transcript, not only the annotation values had to be compared but also the temporal overlap of the inserted tags. To account for this, we calculated the modified Cohen's kappa [63] using the implementation provided by the annotation software ELAN [64, 65]. With the required minimal overlap percentage was set at 61%. We choose this value, following Holle and Rein, for a more "conservative and robust estimation of interrater agreement" [63]. The obtained results, excluding unmatched annotations, yielded a raw agreement of 0.785 and kappa of 0.682. The overall average overlap was 0.658. The achieved values fall into the category of substantial agreement [66, p. 165], which shows that annotators can reliably identify the head movements as head nods and distinguish between different forms of head nods. After calculation of IAR, the annotators proceeded to the full scale annotation of form in the rest of the transcripts separately. The next two steps in the annotation process (annotation for function and turn-taking) were a joint effort of both annotators, that is why no further IAR measures are reported.

**Function.** The second annotation tier was used to assign the functions to the identified head nods. As this study specifically concentrates on two functions of head nods, we differentiated between the following three entries while annotating the function:

- feedback head nods—all head nod occurrences functioning as interactional moves that display a non-uptake of a conversation turn, signal understanding of the information represented by another signer, indicate a prior statement as newsworthy or evaluate the information represented by another person (see the example in Fig 1);

- affirmative head nods—all of the occurrences that signaled an affirmative/positive answer to a question posed within the discourse by the interlocutor (see the example in Fig 2);

- other—this category was used to include all other instances of head nods which were not instances of feedback or affirmative nods.

**Table 1. Tags used for the form annotation.**

| Tag | Description | Number of peak-trough pairs |
|-----|-------------|----------------------------|
| sn | small head nod | one |
| ln | large head nod | one |
| hnn | many small head nods | more that one |
| lnn | many large head nods | more that one |
| mn | one large nod followed by small nod(s) or vice versa | more that one |

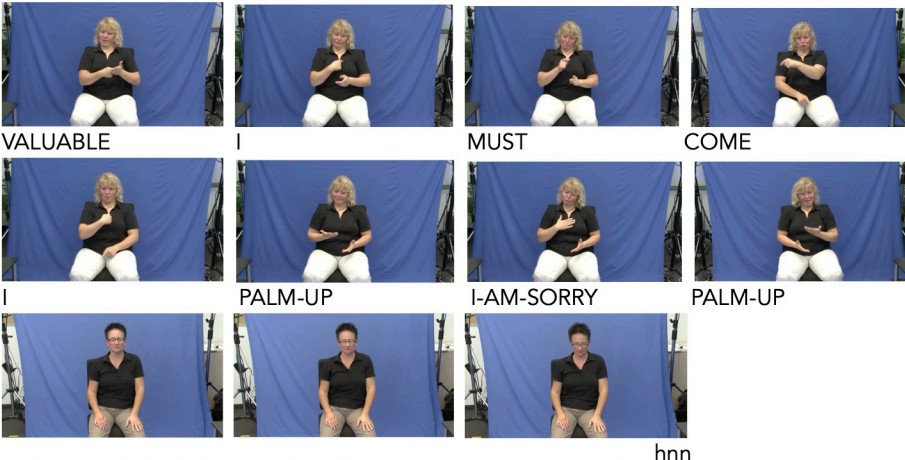

Signer A: I have to go; it means a lot to me. I am sorry.
Signer B: *head nod*

**Fig 1. Feedback nod.** Signer A informs signer B that she is unable to meet with her at a proposed time. Signer B nods her head in order to demonstrate understanding. No manual signs accompany or follow the nod showing that signer B is not willing to take over the conversational turn. This nod has been categorized a feedback nod. Transcript & time-stamps: nue_06_calendar_task; 00:01:57:461—00:01:59:740.

Assigning a function to each identified head nod was not always straightforward and in a small number of cases (<5 cases in the whole dataset) differentiating between affirmation and feedback function was found to not be clear-cut. Our assignment of head nod functions was governed by the following guideline: when a polar question preceded the head nod, we interpreted a head nod as affirmative. In instances where no preceding question was posed, yet a head nod displayed a reaction to a statement, the head movement was labeled as a feedback nod.

**Turn-taking behavior.** In the last annotation step we coded for turn-taking behavior. In coding a non-uptake or a claim of a conversation turn we drew on conversation analytic research on social interaction [67, 68]. Following Levinson, we define turns as "the units of

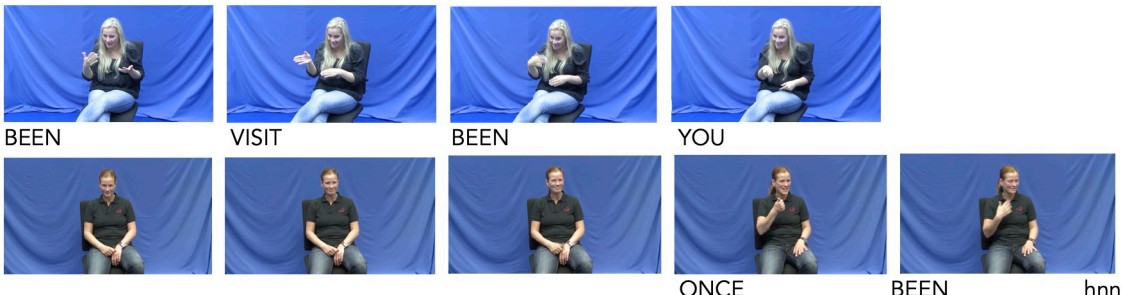

Signer B: Have you ever been there?
Signer A: Yes, I have been there once.

**Fig 2. Affirmation nod.** Signer B asks signer A a direct question, to which signer A produces a positive response. The affirmative head movement precedes the manual signs forming the response and then spreads only the entire articulated clause. Transcript & time-stamps: ber_12_regional_specialities; 00:08:50:160—00:08:52:560.

conversational communication, expressing a speech act" [69]. Turn-taking, thus, is a generic organization of conversation which aims at achieving the goal of "one at a time while speaker change recurs" [67, p. 726] [70]. Turn-taking organization describes the sets of practices speakers use to construct and allocate turns [67, 71]. These phenomena are central to all human interactions which involve at least two people using language.

For the present study, a useful distinction of addressee's turn-taking signals proposed by Duncan [72] appears to be relevant. He distinguishes two following mechanisms of turn-taking used by the addressee: (1) backchannel signals which do not constitute a turn or claim for a turn and (2) turn-claiming signals. Applying this distinction to American Sign Language (ASL), Baker [55] shows, among other things, that backchannels are marked by nodding, and turn-claiming signals are marked by an increase in size and quantity of the head movement. The increase in size and quantity of head nodding is referred to as "highly effective and intense turn-claiming signal" which forces most signers (or speakers) to yield their turn when confronted with it [55].

The two functions of head nods analyzed in this study are claimed to be fundamentally different with regard to turn-taking behavior. Feedback head nods belong to the responsive items which in most of the cases do not constitute a turn or claim for a turn. In contrast, all affirmative head nods are answers to questions and are expected to claim a turn. We aim to test whether this definition holds true in naturalistic conversation. To achieve this, we employed additional coding to examine the two turn-taking mechanisms proposed by Duncan [72]. We utilize a slightly different terminology in this study ('passive recipiency' is used for backchannel signals and 'turn initialization' is used for turn-claiming signals), aligning with more prevalent terms found in the literature, e.g. [73, 74].

All of the head nod occurrences annotated for form and function (feedback vs. affirmation) in the previous annotation steps were now tagged with respect to their turn-taking behavior. We differentiated between head nods signaling a non-uptake of a conversational turn (passive recipiency) and those signaling the signer's intention to start a turn of their own (turn initialization). The head nods were thus consequently classified based on this differentiation:

- passive recipiency (PR): when a signer produces a head nod without claiming the floor by simply signaling to the other signer that they may continue. The passive recipiency nods may be additionally accompanied by a manual sign (e.g. JA—'yes', GENAU—'exactly', STIMMT —'correct'),

- turn initialization (TI): when a signer produces a head nod and takes the conversational floor right afterwards (i.e. a turn transition takes place within the 300 ms threshold).

## Data

**Source material.** The DGS Corpus is an annotated reference corpus of German Sign Language, 50 hours of which have been made publicly available as the Public DGS Corpus [19]. Its 330 participants use DGS as their primary language of daily life and come from various regions of Germany [75]. The DGS content analyzed in this paper (and all shown video images) are drawn from release 3 of MY DGS—annotated [19, 76], the research dataset that provides Public DGS Corpus recordings with full sign annotations and translations in German and English. Participants provided informed consent for the use of their video materials in linguistic research and publications [77]. The dataset covers a variety of text-types, almost all of which are dyadic in nature.

**Table 2. The MY DGS—Annotated transcripts annotated in this study.**

| Annotation | Transcript | Age | Gender | Length | DOI |
|---|---|---|---|---|---|
| Full | ber_07_deaf_events | 31–45 | mm | 00:09:28 | https://doi.org/kz89 |
| | fra_01_deaf_events | 18–30 | mm | 00:03:42 | https://doi.org/kz9b |
| | goe_08_discussion | 18–30 | ff | 00:16:27 | https://doi.org/kz86 |
| | koe_04_free_conversation | 18–30 | mf | 00:18:10 | https://doi.org/kz87 |
| | nue_06_calendar_task | 31–45 | ff | 00:04:15 | https://doi.org/kz88 |
| Spot | ber_01_deaf_experience_A | 18–30 | mm | 00:05:36 | https://doi.org/kz9c |
| | ber_01_deaf_experience_B | 18–30 | mm | 00:03:02 | https://doi.org/kz9r |
| | ber_12_regional_specialities | 31–45 | ff | 00:19:43 | https://doi.org/kz9g |
| | fra_05_free_conversation | 46–60 | mf | 00:15:11 | https://doi.org/kz9q |
| | hh_02_discussion | 31–45 | ff | 00:18:21 | https://doi.org/kz9n |
| | hh_04_warning_signs | 46–60 | mm | 00:01:30 | https://doi.org/kz9j |
| | koe_10_free_conversation | 18–30 | mm | 00:14:22 | https://doi.org/kz9d |
| | koe_20_deaf_events | 31–45 | mf | 00:05:07 | https://doi.org/kz9h |
| | lei_11_regional_specialities | 18–45 | ff | 00:23:02 | https://doi.org/kz9p |
| | mst_08_deaf_events | 61+ | ff | 00:09:00 | https://doi.org/kz9k |
| | mue_02_deaf_events | 18–45 | mm | 00:08:49 | https://doi.org/kz9m |
| | stu_12_deaf_events | 46–60 | ff | 00:06:32 | https://doi.org/kz9f |
| | stu_18_deaf_experience | 18–30 | mm | 00:08:11 | https://doi.org/kz9s |

In addition to translations and gloss transcriptions, each public recording is also provided with pose estimation data [78]. This data represents outputs generated by OpenPose and then post-processed to correct errors related to misrecognition of (nonexistent) additional persons, parts of one person being detected as multiple people, filtering out accidental appearances of the other participant (e.g. their hand reaching high enough to appear in the opposite front-facing recording) and recovery of frames dropped during recognition.

**Annotated material.** The hypotheses described in the section *The current study* are tested based on the analysis of two data samples drawn from *MY DGS—annotated*. The transcripts included in each sample are identified in Table 2. In order to study the head nods in naturalistic language use, we have chosen only transcripts whose task elicits conversational data. We assume nodding to be such a pervasive phenomenon that we did not control for region, age or gender of the participants. As we did not anticipate variance other than the one based on the personal signing style, we include many individual participants and text-types in both samples.

Both samples contain spontaneous and semi-spontaneous conversational data [79]. Spontaneous data come from free conversations in which participants were asked to talk about a topic of their choice while the moderator was absent in the recording room. Semi-spontaneous data belong to two categories: conversational data and elicited data. Semi-spontaneous conversational data come from conversational tasks in which participants were asked to talk about a chosen predefined topic (e.g. to discuss their experiences as a deaf person, to talk about a given deaf event or to talk about issues important for the deaf community like interpreting or bilingual education). Elicited data come from fully-elicited tasks, in which participants were asked to complete a given exercise together. In our data sample, we analyzed three types of elicited tasks (which do not represent all types of elicited tasks present in the DGS Corpus): (1) calendar task (both participants were shown a calendar sheet with various appointments and were asked to schedule a joint meeting); (2) regional specialties (both participants were asked to talk about the specialties of the region they live in); (3) warning and prohibition signs (participants

were asked to figure out and discuss together what the unusual warning and prohibition signs might mean).

The two annotated data sets differ in the number of annotations. The five texts constituting the first data sample were fully annotated for head nodding. In each of the transcript we continuously annotated the video material coming from both signers, which adds up to 104 minutes of annotated material. The amount of annotated material is double that of the transcripts' length, as material coming from each of the two signers simultaneously was annotated separately. In this sample a total of 7826 manual signs is produced (and pre-annotated prior to the start of the present study).

After the quantitative analysis of tags in the first data set we noticed a large imbalance between the number of tokens belonging to the two functional categories (see section *Annotation*), with many feedback tokens (414 cases) and very few affirmation tokens (21 cases). To ensure that the final results contained sufficient numbers of items for both categories we performed an additional round of spot annotation to single out more occurrences of affirmation nods. As our study is concerned with the phonetic differences between types of head nods, rather than their frequency distribution, his methodological approach was considered appropriate. We inspected additional 13 dyadic files which constituted the second data sample. As only the instances of affirmative head nods were singled out and annotated in those files, they are referred to here as spot annotations. All of the spot annotated files lasted 138 minutes and 24 seconds and yielded 62 additional occurrences of affirmative head nods. Taken together, we analyzed 242 minutes and 24 seconds of video material from the DGS Corpus, i.e. approximately 4 hours of naturalistic data.

## Pose estimation

Automatic recognition and classification tools have the potential to support and speed up labor-intensive manual annotation in multimodal communication research. While historically most such tools focused on acoustic signal processing, recent developments in computer vision research have opened up new possibilities in the visual domain [80]. Central to these developments is the emergence of reliable pose estimation, the automatic determination of body part locations in images.

The most impactful pose estimation tools in multimodal communication research are OpenPose [13] and OpenFace [14]. The latter has been described to be less robust to capture head pose and to necessitate a considerable amount of manual verification [81]. It is worth mentioning, however, that OpenFace was not compared with OpenPose, but with MediaPipe and 3DDFA. OpenPose lacks direct head rotation measurements, rendering it challenging to make direct comparisons with other tools.

Pose estimation tools provide information on the location of individual body parts in an image, allowing the tracking of their movement across a video recording. The results are similar to those of motion capture, but are achieved through image recognition instead of the use of specialized mocap suits, making it possible to apply them to existing recordings that did not employ any specialized equipment at the time of recording.

A limitation of pose estimation is its reliance on two-dimensional images lacking depth information, resulting in two-dimensional output. While some configurations of the aforementioned tools can also predict depth, this information is usually considerably less precise than that of the other dimensions [82]. Furthermore, neither pose estimation nor motion capture provide any interpretation of what the tracked motion signifies, e.g. while they may track the vertical motion of a head, they offer no judgment as to whether this motion would be perceived by a human as nodding.

**Fig 3. Visualisation of head nod analysis based on pose estimation keypoints.** The source video (left) is overlaid with the OpenPose keypoints used for the calculations. On the line graph (upper right) the line represents the vertical motion of the nose relative to body position with crosses indicating peaks and troughs and light blue boxes indicating durations manually labeled as head nods. The frequency spectrogram of nod movements (lower right) was included as an experimental visualization, but excluded from the study as it was found to only be reliable for very regular larger nods. Source video origin: MY DGS—annotated [76].

Using pose estimation for the linguistic analysis of signed languages has been gaining momentum in recent years. A number of researchers applied computer vision to measure the properties of manual actions used by signers, e.g. [10, 12, 83–86] among others.

As mentioned in the section *Introduction*, we follow the approach of other authors who have successfully used pose information in analyses of other non-manual markers in sign languages [15–17] and showed that combining the manual annotation of video recordings with the computation of measurements based on pose estimates appears to be the most suitable for linguistic analysis of non-manuals so far. Therefore, we analyze the phonetics of head nods using the same approach.

## Calculating phonetic properties

To test our predictions described in section *The current study*, we calculate phonetic properties based on our head nod annotations (see section *Annotation*) and the OpenPose pose information provided with the DGS Corpus transcripts. A visualization of this analysis is shown in Fig 3. To account for differences in person size and distance from the camera, the body point coordinates were normalized according to the distance between hip and collarbone of each respective participant. We track nodding by observing the motion of the nose relative to the position of the center of the collarbone. Hip and collarbone keypoints are taken from the general body model [13] (keypoints 1 and 8 in left image of Fig 4), as nose keypoint we take the tip of the nose of the specialized face model [87] (keypoint 30 in right image of Fig 4). To account for imprecisions in keypoint estimation, we apply a second order Butterworth filter [88] with a zero-phase lowpass cutoff of 20, using the filter implementation from *SciPy* [89] and adapting code by Wim Pouw [90]. The source code for all calculations is part of the public dataset.

In this paper, we focus our analyses on the three following phonetic properties of head nods, as these are the ones relevant for our first hypothesis (see section *The current study*):

**Duration.**   The time elapsed between beginning and end of a head nod. This information is derived directly from the manual annotations and does not rely on pose information.

**Velocity.**   The speed of the nodding motion, measuring how much distance the nose travels during a nod per second, averaged over its duration. Velocity is defined as the derivative of distance over time, which we determine by computing the absolute vertical distance of the nose between adjacent frames for each frame pair of the nod duration and then determining the mean average value of all pairs.

**Maximal amplitude.**   The magnitude of the nodding motion, taken to be the distance between the highest and lowest peak of the nod. In this we follow Chizhikova and Kimmelman

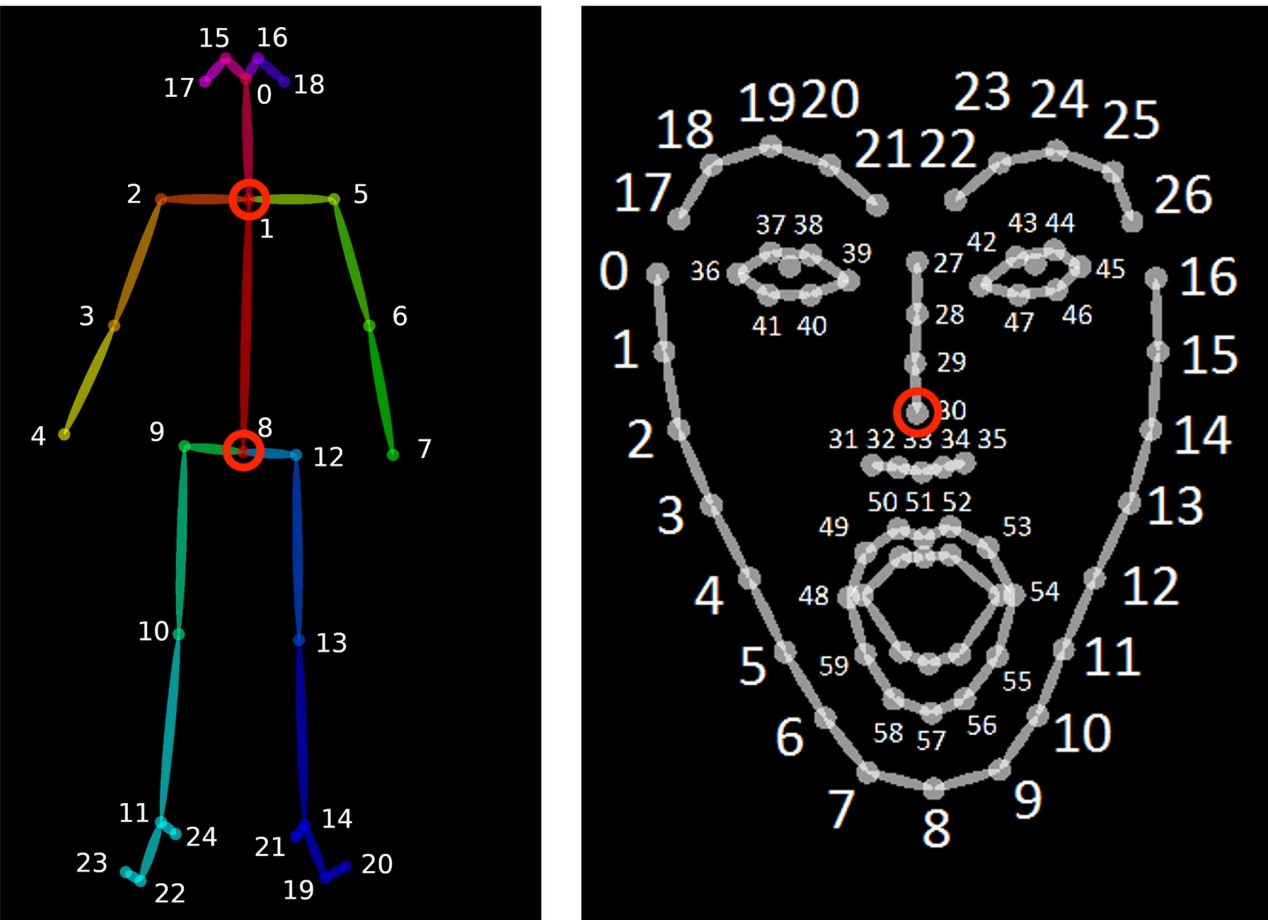

**Fig 4. Body and face keypoint sets determined by OpenPose.** The keypoints used in this study are circled in red. Image source (highlights ours): https://github.com/CMU-Perceptual-Computing-Lab/openpose.

[17]. Peaks are calculated using the `find_peaks()` method of the Python library *SciPy* [89], requiring a minimum prominence of 0.1 and minimum distance between peaks of 5 frames (i.e. 100 milliseconds) to filter out noise caused by the imprecisions in pose estimates. This measure can be misleading in the rare case that the basic head position changes during the nodding motion, e.g. a participant looks up while nodding, resulting in a greater distance between lowest and highest point than was achieved between any two adjacent peak and trough. This could be avoided by basing computations on adjacent peaks, however, due to the error margins of pose estimates, it is difficult to determine whether all potential intermediate peaks are real or prediction noise. For the same reason we do not include peak frequency as a measurement in our study. We also considered (topographic) prominence as an alternative to amplitude, as it is designed to ignore minor peaks in a contour. However, we found the observed trends to be the same for both measures and therefore decided to stick with maximal amplitude, as it has already been established in relevant prior work.

## Statistical analysis

Below, we present results from binomial generalized linear mixed models (GLMMs) using the lme4 package [91]. Our dependent variable was the proportion of affirmative head nods. We

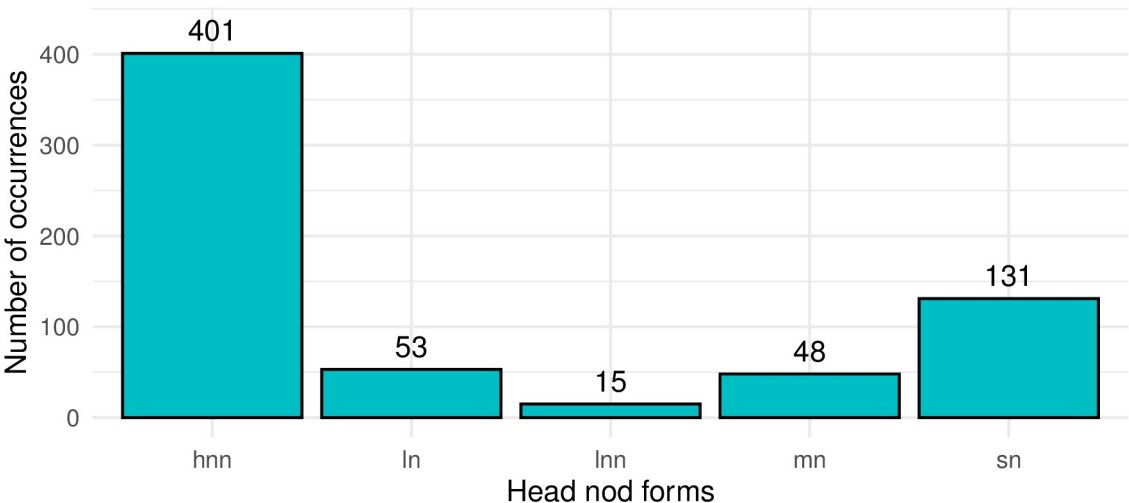

**Fig 5. Distribution of head nod forms in the analysed dataset (n = 648).** hnn = many small head nods; ln = large single nod; lnn = many large nods; mn = mixed nod (e.g. one large and many small nods), sn = small single nod.

included the fixed effects of velocity, maximal amplitude, and duration, and a random effect for speaker. Since affirmations were spot-annotated relatively more often, the number of observations per speaker was not equal. However, we decided not to control for the number of observations per speaker, because we were interested in the effect of the fixed effects on the probability of an affirmative head nod, and not on the number of affirmative head nods. We z-scored all three effects and then inspected them for outliers and multicollinearity. We observed a few extreme nods, e.g. a nod of 13.02 seconds whereas the mean nodding duration was 1.56 seconds (considering feedback and affirmation nods only), which would potentially have a large influence on the modeling outcomes. To deal with such outlier values in the logistic regression analysis, we decided to remove outliers that were more than 5 times the IQR away from the overall quartiles, which can be considered very conservative. This resulted in the removal of 8 of the 497 observations. Furthermore, we observed that velocity and amplitude were correlated with each other (r = .66, see Fig 11). We used R version 4.3.0 [92]. The analysis code is part of our public data (see section S1 File).

## Results

In total, we analyzed 648 occurrences of head nods in the data sample. Our annotation results reveal that multiple small head nods was the most common type of head nodding in the inspected data (this distribution would have remained consistent even upon scrutinizing the initial dataset independently of the spot annotated data). This finding is consistent with earlier research on the use of head nod in languages belonging to both modalities [4, 6]. A summary of the distribution of form and function tags is given in Figs 5 and 6 respectively. The total number of multiple small nods in the data was 401, which constitutes 62% of all analyzed head nods. The second most numerous category was single small nods, accounting for 131 occurrences and 20% of all analyzed head nods. Larger head nods usually consist of one peak-trough pair (53 instances), instances of multiple large nods are very rare (15 instances).

As can be seen in Fig 6, feedback nods constitute the majority (64%, n = 414) of identified cases which appears to align with earlier observations for head nod functions in spoken and signed languages [6, 20]. Affirmative function has been assigned to only 13% (n = 83) of

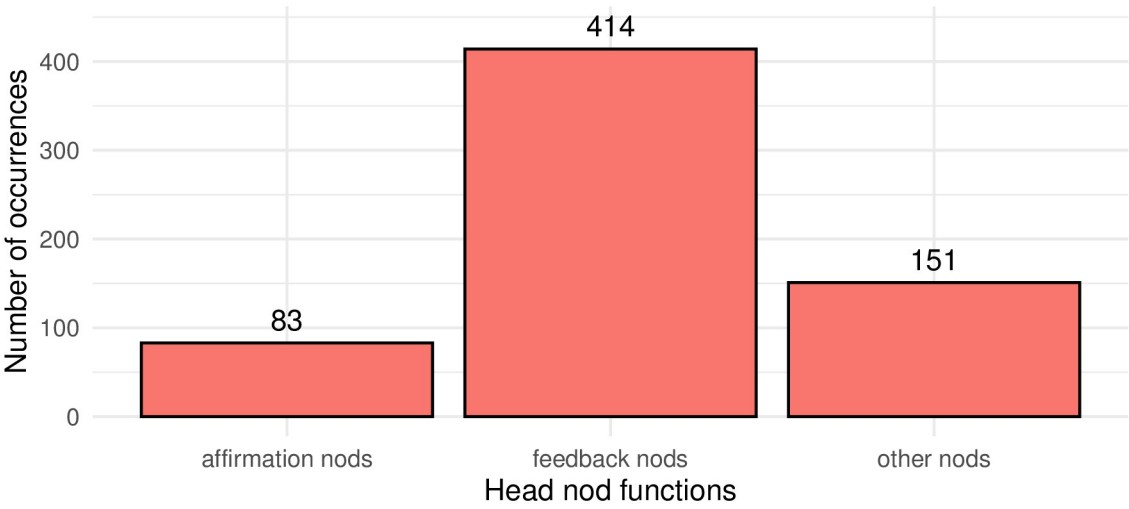

**Fig 6. Distribution of head nod functions in the analysed dataset (n = 648).**

occurrences in the data sample. 151 instances (23%) of nods annotated with the tag 'other' are excluded from the following analyses and are left for future research. The percentages provided above serve to furnish the reader with an overview of the nodding behaviors under analysis within our selected dataset. It is imperative to note that these figures do not signify natural occurrences, as the dataset was deliberately expanded to mitigate disparities in the incidence of the two nod types.

Investigating the form-function pairing of feedback and affirmation head nods (see Fig 7), we find that both functions are predominantly carried by the multiple small head nods, although multiple small head nods have a higher tendency to generate the feedback function rather than affirmations. Another difference we observe between the two groups is that when large nods are produced (whether single or multiple), they tend to typically signal affirmation.

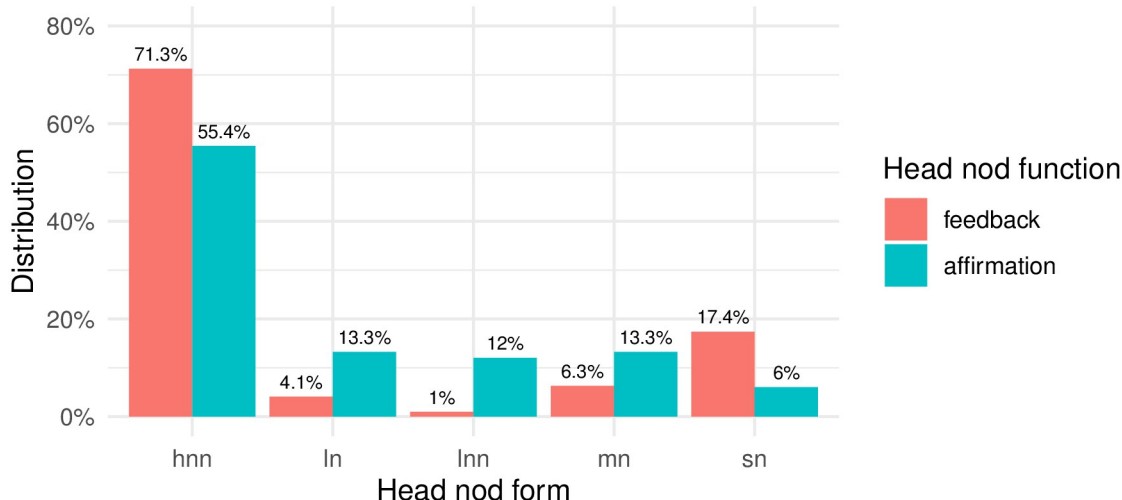

**Fig 7. Percentages of different form and function types in the analyzed sample.**

**Table 3. Average duration of head nods in DGS.**

| Tag | Mean duration (sec) | Median duration (sec) | Number of occurrences |
|---|---|---|---|
| sn | 0.61 | 0.78 | 131 |
| ln | 0.79 | 0.90 | 53 |
| hnn | 1.69 | 1.14 | 401 |
| lnn | 1.93 | 1.96 | 15 |
| mn | 2.31 | 1.92 | 48 |
| Overall | 1.45 | 1.04 | 648 |

Relying solely on manual annotations, our findings indicate that feedback nods tend to be produced with a small single or multiple nods, while affirmation nods are more frequently associated with larger nods than feedback nods.

## Basic phonetic properties of DGS head nod

The average duration of a standard DGS head nod was 1.45 seconds (see Table 3). The duration of a head nod movement ranged from 0.04 to 13 seconds. When comparing these results to the patterns previously reported for FinSL [6], we observe some similarities and some differences. The duration for single nods in the FinSL data (1.3 seconds on average) was slightly higher than the average duration of DGS single small (0.61 sec) or large nods (0.79 sec). The average of 2.1 seconds for FinSL multiple nods is more comparable with the mean duration of mixed head nods and large head nods in DGS (see mn and lnn in Table 3). In Table 3 we report also median duration in order not to get distorted by extreme outliers.

On average, each head nod movement consisted of 2.03 peaks (or 1.43 peaks/sec), although this value should be taken with caution, as despite smoothing, pose recognition noise may account for some of these (presumed) peaks. As we do not have information on participant height or exact camera-to-participant distance, we calculate amplitude and velocity by taking the video pixel distance and normalizing over the participant's body height (hip-to-collarbone distance) to compensate for variations in person size. We refer to this unit of distances as normalized pixels (npix). These values allow us to compare nods within our own dataset, but unfortunately they are not suitable for comparing with other publications. Within our data, however, we can observe that pose-based amplitude and velocity calculations align with human annotations regarding differences between large and small nods. While there is still a lot of individual variation, the average small nod (median amplitude 0.023 npix, velocity 0.22 npix) is considerably smaller and somewhat slower than the average large nod (median amplitude 0.087 npix, velocity 0.38 npix).

## Differences between affirmative and feedback head nods

For the quantitative analysis of the phonetic properties of affirmation and feedback head nods, we focus on the three features: duration, velocity and amplitude as described in the section *Calculating phonetic properties*. We find significant differences in some phonetic properties of head nods used in affirmative and feedback functions and observe different patterns of the co-occurrence with the manual elements. Out of those 648 tags, which served as a basis for computing phonetic properties of a 'standard' head nod in DGS, 151 were annotated as having a function 'other' and are excluded from further analysis. In the following we focus on the remaining 497 annotations of head nods.

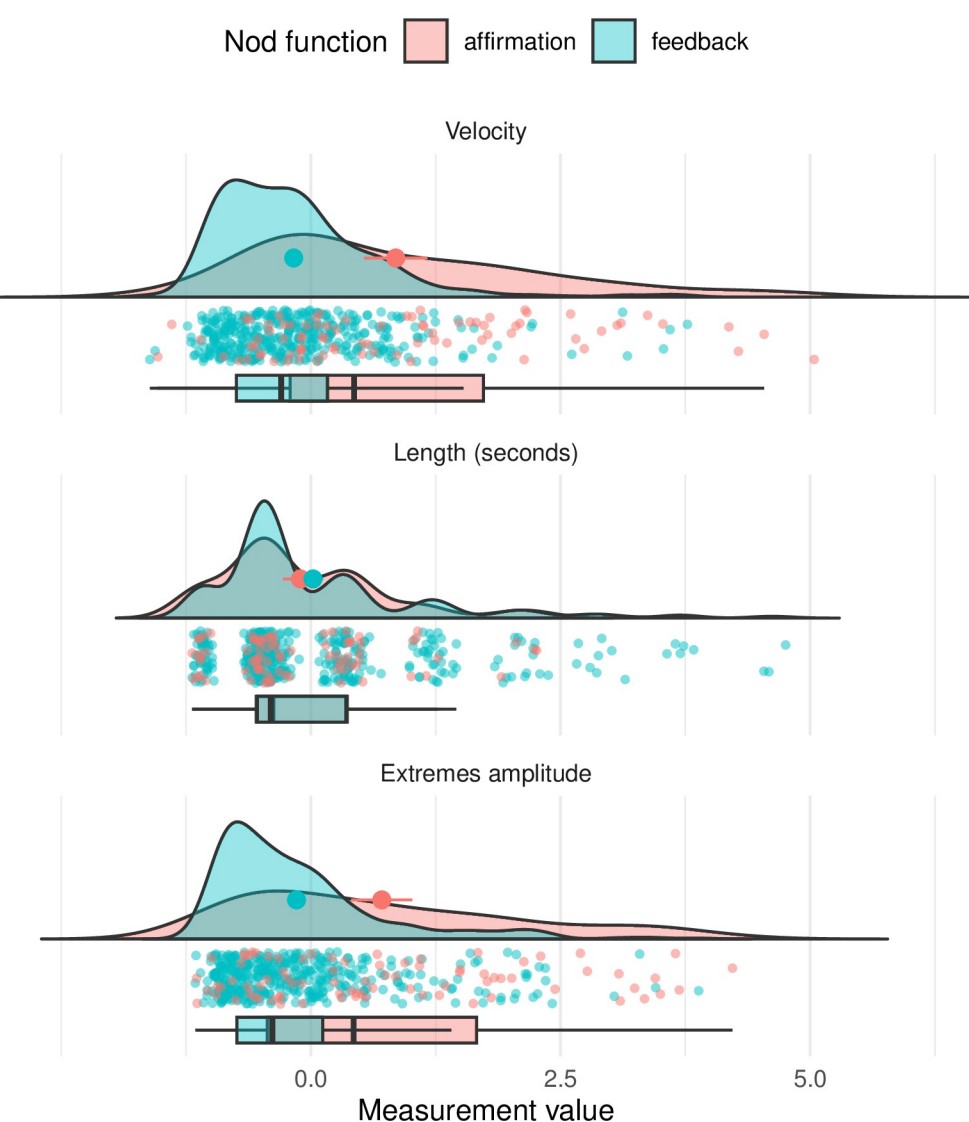

**Fig 8. The distributions of the three variables, showing their density, their outliers, and box plots.** The plots include outliers that we later removed from the analysis (points further away than 5 * IQR from the box plots).

**Phonetic properties.** The duration of the labeled head nod was annotated manually in ELAN, therefore computer vision tools were not required to calculate this measure. For the velocity and maximal amplitude measurements we relied on the OpenPose pose information (see section *Pose estimation*). We assumed in our hypotheses (see section *The current study*) that affirmative nods will be larger in amplitude, shorter in duration and faster in their production than head nods which signal feedback in interaction.

For the quantitative analysis, we compared the mean measurements of affirmative and feedback nods. The distribution of velocity, duration and extremes amplitude are shown in Fig 8 and tests for statistical significance of these values between nod functions are given in Table 4. Regarding duration, feedback nods (1.6 sec ± 1.35 sec; median = 1.08 sec) exhibit a larger number of particularly long outliers than affirmative nods (1.45 sec ± 0.99 sec; median = 1.08 sec),

**Table 4. T-tests and Wilcoxon tests comparing three phonetic properties between feedback and affirmation nods.**

| Variable | t-test (sig.) | | Wilcoxon test (sig.) | |
| --- | --- | --- | --- | --- |
| Amplitude | 5.37 | *** | 22889.00 | *** |
| Duration | -1.23 | | 16123.50 | |
| Velocity | 6.25 | *** | 24456.00 | *** |

but their median duration is identical and differences are not statistically significant, contradicting our hypothesis for duration.

In line with our predictions, we observe significant differences in the velocity and amplitude measurements. Affirmative head nods are larger in amplitude and faster in their production than feedback nods. This would indicate support of our hypotheses for velocity and amplitude, however, we perform additional modeling to confirm the importance of these attributes.

Model comparison (using GLMMs, see section *Statistical analysis*) showed that head nod velocity significantly improved a baseline model that only contained a random effect for speaker ($\chi^2$ = 8.026, p = .0046), and that head nod duration did not ($\chi^2$ = .5944, p = .4407). Also, velocity improved a model that already contained duration ($\chi^2$ = 8.1317, p = .0044), but not the other way around ($\chi^2$ = .70, p = .4027). Amplitude improved the baseline model by a nonsignificant margin compared to velocity ($\chi^2$ = 2.7201, p = .0990), but improved the model that already contained duration, although less than velocity did ($\chi^2$ = 4.9069, p = .0267). Likewise, amplitude did not improve a model with velocity ($\chi^2$ = .0031, p = .9554), but velocity improved a model with amplitude ($\chi^2$ = 5.3090, p = .0212). A model with all three variables did never improve a model significantly when either duration or amplitude ($\chi^2$ = 3.4261, p = .0641) duration and velocity ($\chi^2$ = 0.2014, p = 6536), or velocity and amplitude ($\chi^2$ = .8984, 3432), respectively. Based on these comparisons, we conclude that a model with only velocity as a predictor represents the data most parsimoniously (see also Fig 9), compared to a model with all three predictors. Furthermore, the graph in Fig 10 shows the adjustments for individual signers based on a model with velocity only. This figure gives an overview of the skewed distribution of data points across signers due to spot annotation and also shows that the model estimates an adjustment to the velocity effect for every unique signer depending on the data available.

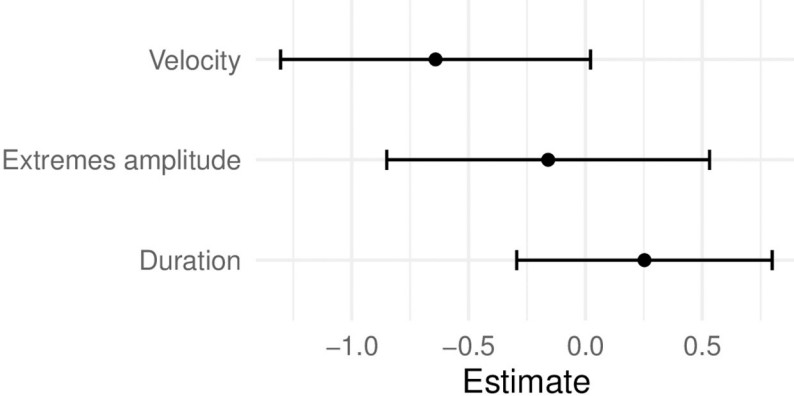

**Fig 9. Coefficients and standard errors based on a logistic regression model with all three predictors.**

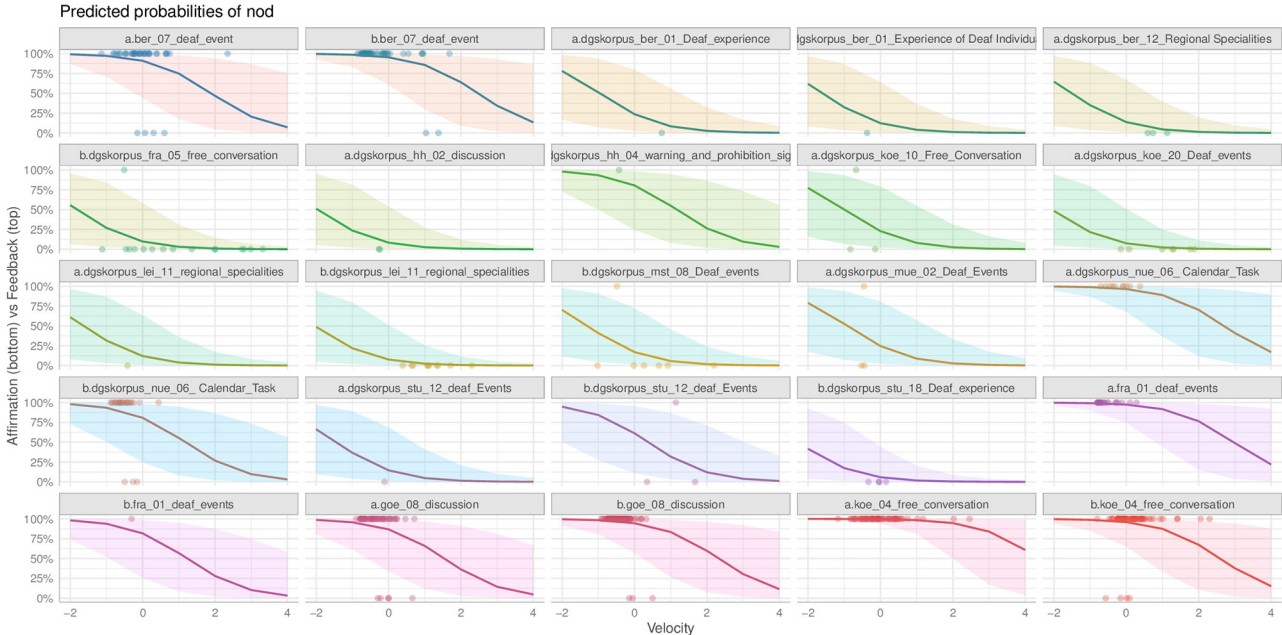

**Fig 10. Predicted value for every signer.**

These regression results present a more detailed picture compared to the separate tests above that show a significant difference for both velocity and amplitude. Amplitude does not add anything on top of velocity, and velocity improves the model more than amplitude does. This pattern can be explained by the high correlation between the two (r = 0.7445, with a 95% CI between 0.7021, 0.7816), with a stronger correlation for the affirmation nods (0.8182, 95% CI: 0.7310, 0.8791), compared to the feedback nods (0.6457, 95% CI: 0.5850, 0.6991), as shown in Fig 11.

Finally, the estimated effect size for the velocity difference between feedback and affirmation was -1.1046 (95% CI: -1.3525, -0.8567), which can be considered a large difference. Given its skewness, we also computed a bootstrap distribution of median differences, which resulted in a very similar effect size (-1.1005, 95% CI: -1.3497, -0.3618).

**Co-occurrence with manual items.**   The second hypothesis in this study (see section *The current study*) was concerned with the overlap of head nods with other cues in interaction, such as lexical and non-lexical manual (i.e. articulated by the signers' hands) elements. To offer a cohesive overview of manual items that align with affirmative and feedback head nods in DGS we combine the annotations already present in the Public DGS Corpus with the annotations generated for the purposes of the current investigation. The pre-existing annotations included glosses for both the lexical signs and gestures produced by either or both hands of each of the participants. Tags added in the course of the present study, which were inserted on the parallel tiers, let us calculate the overlap of the feedback and affirmative nods with the different kinds of manual occurrences in ELAN [64, 65].

We apply the division of manual cues into lexical and gestural items as already present in the DGS Corpus and reflected in the glossing system [93]. We observe substantial differences between the two groups of head nods in the co-occurrence with manual items (see Figs 12 and 13). While the vast majority of feedback nods are articulated without any manual element (65%, see Fig 12), 87% of all affirmative head nods were produced together with a manual item

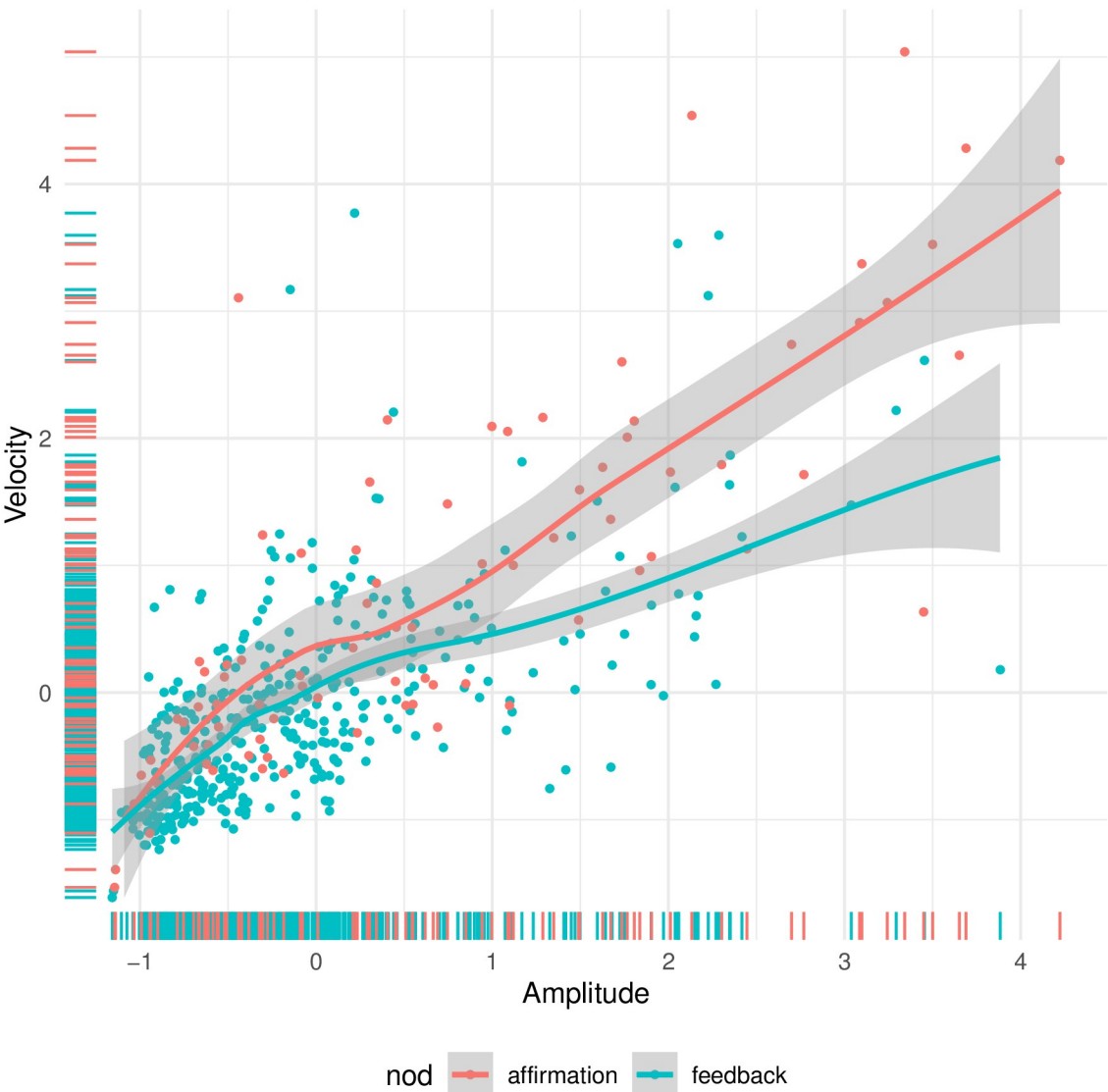

**Fig 11. Correlation plot for velocity and amplitude in order to inspect their colinearity.**

(see Fig 13). With these findings we confirm our second hypothesis that affirmation nods co-occur with manual items more often than feedback nods do.

Typical manual lexical signs which accompanied feedback head nods were INDEX, YES (JA), AGREED (STIMMT), GOOD (GUT), OKAY (OKAY), NO (NEIN), BUT (ABER), EXAMPLE (BEISPIEL) and others. The most common gestural items that co-occurred with the feedback nods were palm-up gestures (28 cases out of 40 manual gestures co-articulated with feedback nods). Affirmative head nods usually co-occurred with such lexical elements as YES (JA), AGREED (STIMMT), OK, EQUAL (GLEICH), CAN (KÖNNEN), ALSO (AUCH) and others. As for gestural items accompanying affirmative head nods, here also palm-up gestures were the most common (8 cases out of 11 cases of manual gestures co-articulated with affirmation nods). This is not a surprising finding, as palm-ups are among the most frequently occurring manual elements in a number of sign languages.

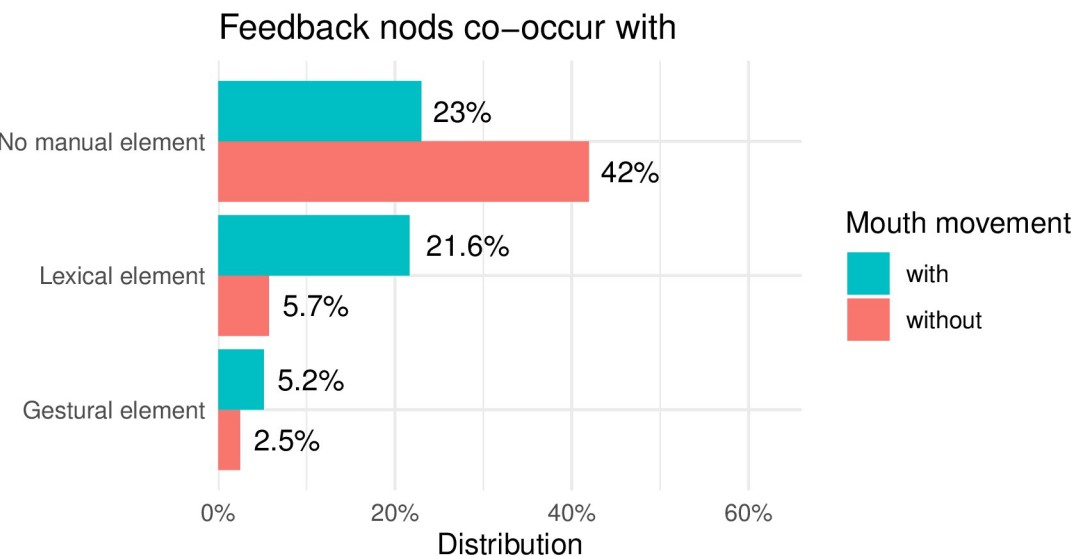

**Fig 12. Overlap or lack thereof of feedback nods with manual items and mouth movements.**

Additionally, the palm-up gesture family consist of a number of forms, yielding the gesture not only very frequent, but also highly multifunctional [94].

During the annotation process we noticed that head nods of both types often co-occurred with different types of mouth movements even in the absence of a manual element. Therefore, we decided to include the annotation tier for mouth movements (already present in the Public DGS Corpus) in the current analysis. Each head nod was inspected for its co-articulation with any type of mouth movement—either mouthing or mouth gesture as these two mouth movement types are annotated on the same tier in the DGS Corpus. Mouthings are transcribed in

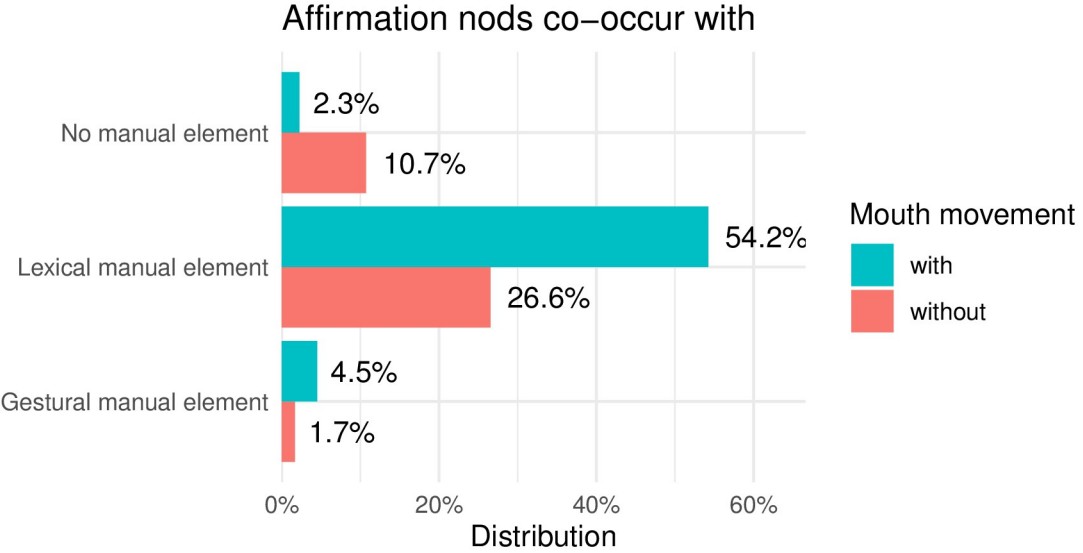

**Fig 13. Overlap or lack thereof of affirmation nods with manual items and mouth movements.**

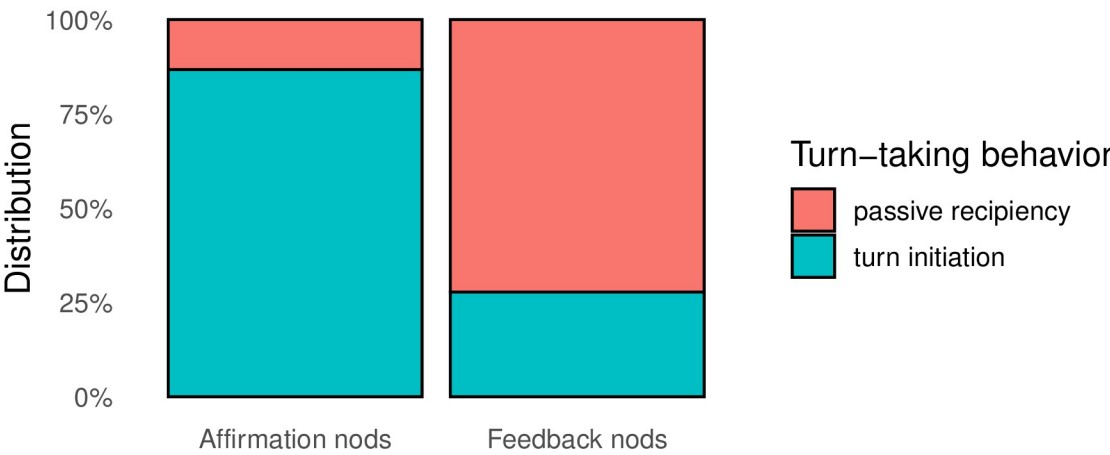

**Fig 14. Turn-taking behavior and function of the nods.**

full, while all mouth gestures are annotated with a single tag ([MG]) and are not differentiated in further subtypes.

As can be seen in Figs 12 and 13, in instances where head nods co-occur with lexical signs, a concurrent presence of mouth movement is also observed. Most of these mouth movement are "standalone non-redundant mouthings" [52], meaning that mouthings also accompany head nods even if manual signs are absent. The most typical cases of mouthings are 'stimmt', 'ja' and 'klar'. The data presented in Figs 12 and 13 are based entirely on annotations from the Public DGS Corpus. However, these annotations are likely incomplete, as the focus of annotation in the corpus lay on the active signer, foregoing annotation of feedback signals and interjections in many cases.

**Turn-taking behavior.** As the result of the third annotation round (see section *Turn-taking behavior*) we have identified 187 nods which initiated a turn and 310 nods produced in passive recipiency. Our primary focus was, however, to explore the potential distinction in turn-taking behavior among head nods associated with two different functions (see Fig 14). As expected, a substantial disparity emerged between the two groups. 72.2% of feedback nods signal passive recipiency, indicating the turn continuation of another signer. In contrast, affirmation nods primarily signal turn initiation resulting in a turn shift (86.7%). Interestingly, there is a number of cases that go against this trend, as 27.8% of all feedback nods do initiate a turn and a small number of affirmative nods do not signal a turn claim (13.3%) (see Fig 14).

A direct comparison between feedback nods initiating a turn (n=115) and the feedback nods which signal passive recipiency (n=299) revealed significant differences in velocity and duration (see Table 5). Passive recipiency head nods are longer in duration and slower in their production of the movement than those feedback nods initiating a turn.

**Table 5. T-tests and Wilcoxon tests comparing three phonetic properties between feedback nods that do and do not claim a turn.**

| Variable | t-test (sig.) | | Wilcoxon test (sig.) | |
|---|---|---|---|---|
| Amplitude | 0.61 | | 18089.00 | |
| Duration | -4.04 | *** | 12225.00 | *** |
| Velocity | 4.58 | *** | 22289.00 | *** |

## Discussion & outlook

### Main findings

The major goal of this study was to examine whether head nods serving different functions vary in their phonetic characteristics. We approached this by analyzing natural dyadic conversations, one of the most common forms of interaction [95], from the Public DGS Corpus, using a novel method of combining manual annotations in ELAN and measurements extracted from a CV tool, OpenPose.

This study specifically concentrated on feedback and affirmation functions of head nods, which have received little attention in existing sign language literature so far. An important finding of this study is that head nods functioning as affirmative responses differ in their phonetic properties and their alignment with manual elements from those head nods, which signal feedback in interaction. Our main finding is that feedback nods are on average slower and smaller than affirmation nods, and they are commonly produced without a co-occurring manual element. However, they frequently co-occur with other non-manual cues, such as mouthing.

In line with our hypothesis, our results reveal significant differences in velocity and amplitude between the feedback and affirmative head nods. Velocity as a predictor provides the most parsimonious representation of the data, which can be explained by the correlation between velocity and maximal amplitude (see Fig 11). The positive correlation between the velocity of a head nod and the amplitude of the movement is likely attributed to inertial principles, as a larger movement requires more force and speed to execute compared to a smaller movement. Consistent with this observation, in the analyzed DGS data, we observe that a larger amplitude head nod tends to be associated with a higher velocity, particularly in functions related to feedback and affirmation, while a smaller amplitude nod tends to have a lower velocity. Contrary to our hypothesis, our findings show no significant difference in duration between feedback and affirmation head nods. However, we do observe a notable discrepancy in duration between feedback nods that do and do not initiate a conversational turn. This lends support to our explanation pertaining to turn-taking behavior provided below.

### Explanation

We explain the differences in phonetic properties found in feedback and affirmation head nods by the different roles these cues play in the conversational turn-taking system [67, 72]. Our results show that the overwhelming majority of feedback nods identified in the data are passive recipiency signals, which are produced without displaying an intention to claim the conversational turn. Almost 90% of affirmative nods, in contrast, signal initiation of a turn. We interpret our findings as suggesting that the form of a head nod may be adapted to or influenced by its function in the turn management system, manifesting variations in parameters such as velocity and amplitude of the movement. Thus, the signers make use of the non-manual cues, e.g. head nods, in face-to-face interaction in accordance with discourse management strategies. We suggest that this flexibility of non-manual cues in interaction enhances both efficiency and progressivity in communication.

With the increasing availability of sign language corpus data produced in naturalistic settings, there has been an increase in studies examining everyday face-to-face interaction, the primordial setting of human communication [69]. Recent research reveals that human communicative interaction strives for efficiency, clarity, and minimization of breakdown or communicative trouble [57, 96–98]. Turn transitions happen at a very rapid rate [99]. It is important to quickly understand the interlocutor's intended message and to provide a fast

response. Communicators even start planning their own response while still perceiving the previous turn [100]. Non-manual cues have been reported to largely contribute to this fast dynamic and to ensure progressivity in interaction [52, 95, 98, 101–104]. Visual cues appear to be best suited for this, as "the core of social interaction perception is visual" [105, p. 1165].

Our research draws attention to the importance of non-manual cues in the organization of the turn-taking system in sign languages.

Sign languages involve the coordination of multiple active articulators. The hands produce manual signals, while non-manual articulators such as the mouth, eyes, or head concurrently generate non-manual cues. By reducing a unit that usually consists of manual and non-manual activity to only non-manual, such as a head nod, as it is the case with feedback nods, the cognitive load lessens, and the signal becomes more readily and rapidly perceived. Enhancing the visibility of a signal by increasing the size and speed of the head nod or by adding a manual element to it results in a more prominent signal, which is easier to detect and which displays the potential for interrupting the conversational flow. The realization of different head nod forms depending on a conversational function appears to be beneficial for the exchange as it aids more efficiency in language and communication.

Our results show that turn-taking behavior has an impact on some phonetic properties of head nods in DGS. Our findings suggest that when signer displays a non-uptake of a conversational turn, their head nods are smaller and produced slower than the nods which occur in the context of turn initiation. Naturalistic conversational data show that feedback nods while sending a signal of attentiveness and engagement in conversation are found to also initiate a conversational turn in a number of cases. In this case, feedback head nods have both higher velocity which makes them more visible, and are produced faster than those signaling a non-uptake of a conversational turn. The fact that some feedback nods have been found to be followed by a conversational turn despite their definition as being not turn-competitive can be attributed to our lack of differentiation between various subcategories of feedback, such as continuers, acknowledgments, newsmarkers or assessments.

Slow and small head nods produced without manual signals do not disturb the flow of the signer's signing, enable the other to still perceive the movement while signing, and at the same time they let the addressees express their continued attention and participation in interaction. This confirms and amplifies previous observations in the literature, e.g. for spoken English [58] and for ASL [55].

## Limitations of this study

One of the identified limitations relates to the annotation of the affirmative nods. In our data, affirmation nods usually align with a manual response to the question asked by the interlocutor and the nodding might spread from the affirmative particle onto the following constituents, which constitute the overall response (see the example in Fig 2). When annotating head nods, we refrained from segmenting a continuous head movement into distinct head nods, even when they were coarticulated with different manual items. For example, one affirmation nod can spread over a constituent, such as YES I WAS THERE BEFORE. A feedback nod can spread over two or three manual signs, e.g. YES RIGHT I AGREE. We chose a uniform approach to annotate head nods, independent of their alignment with observed manual activities. This decision was made due to the occurrence of diverse manual activities associated with feedback and affirmative nods. The chosen annotation method has yielded affirmation nods with prolonged durations as in the example in Fig 2. This might be the reason for observing no differences in duration between affirmative and feedback nods.

The second limitation is the low number of observations for the affirmative nods we identified, which can influence the reliability of the statistical analysis. In dyadic face-to-face interaction, we identified a substantial frequency of feedback nods, which is contrasted with the scarcity of affirmation nods. Consequently, targeted spot annotations were necessary to identify sufficient numbers of instances of affirmation nods. As a result, the distribution of head nod types is skewed in the overall dataset. This is acceptable for the presented study, which is concerned with the analysis of phonetic properties. However, any future studies relying on the frequency distribution of head nod types should be limited to the subset of fully annotated transcripts. There is also the possibility that the set of spot annotated transcripts differ from the fully annotated set in ways that affect the general head nod characteristics observed in the respective sets, although based on our observations and given that both sets cover similar types of tasks and contain a variety of different participants, we consider this unlikely.

The use of OpenPose pose information for the computation of analyses also introduces certain limitations, especially with regard to the estimation of head nod peaks. The automatic recognition of body part positions as keypoints has limited accuracy, resulting in inconsistencies of keypoint positions between frames, resulting in jitter. To filter out this noise for calculations, smoothing is applied. The estimates of where the peak of a head nod occurs may be thrown off both by recognition jitter as well as by overly aggressive smoothing. Even without these factors, automatic peak detection may not fully match what human coders would perceive as head nod peaks.

An additional limitation of the pose-based calculations is the normalization based on body height. While the body height value was computed based on an average of different frames, differences in posture might still affect the exact calculations.

## Conclusion and future work

With this study we have also shown, based on naturalistic interactional DGS data, that measurements based on automatic pose information do align with human judgments with respect to, e.g. the amplitude and velocity differences between small and large head nods. These measurements can be combined with human classifications to enable quantitative analyses that would be too costly to produce through fully manual annotation efforts. For the current study, we did not compare head nods in DGS to head nods produced by signers of other sign languages and to gestural head nods produced by speakers of German or other spoken languages. Cross-linguistic as well as cross-modal differences in terms of phonetic properties of head nods might be expected and we intend to address this in future research.

## Supporting information

**S1 File.**
(PDF)

## Acknowledgments

We thank Lina Herrmann for the assistance with head nods annotations in one of the DGS files. We also thank our colleague, Vadim Kimmelman, for his invaluable feedback and insightful comments on an earlier version of this manuscript. His input contributed to its refinement.

## Author Contributions

**Conceptualization:** Anastasia Bauer.

**Data curation:** Anastasia Bauer, Anna Kuder, Marc Schulder.

**Formal analysis:** Marc Schulder, Job Schepens.

**Funding acquisition:** Anastasia Bauer.

**Investigation:** Anastasia Bauer, Anna Kuder, Marc Schulder, Job Schepens.

**Methodology:** Anastasia Bauer, Anna Kuder, Marc Schulder, Job Schepens.

**Project administration:** Anastasia Bauer.

**Resources:** Marc Schulder, Job Schepens.

**Software:** Marc Schulder, Job Schepens.

**Supervision:** Anastasia Bauer.

**Validation:** Anna Kuder, Marc Schulder.

**Visualization:** Anastasia Bauer, Marc Schulder, Job Schepens.

**Writing – original draft:** Anastasia Bauer, Anna Kuder.

**Writing – review & editing:** Anastasia Bauer, Anna Kuder, Marc Schulder, Job Schepens.

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
