## [Decision Letter · Decision Letter 0]

27 Feb 2024

PONE-D-24-04524Phonetic differences between affirmative and feedback head nods in German Sign Language (DGS): A pose estimation studyPLOS ONE

Dear Dr. Bauer,

Thank you for submitting your manuscript to PLOS ONE. After careful consideration, we feel that it has merit but does not fully meet PLOS ONE’s publication criteria as it currently stands. Therefore, we invite you to submit a revised version of the manuscript that addresses the points raised during the review process.

We look forward to receiving your revised manuscript.

Kind regards,

Laura Morett

Academic Editor

PLOS ONE

Journal Requirements:

"German Science Foundation (DFG) SPP 2392 Visual Communication (ViCom)"

Please state what role the funders took in the study.  If the funders had no role, please state: ""The funders had no role in study design, data collection and analysis, decision to publish, or preparation of the manuscript."" If this statement is not correct you must amend it as needed. 

"This paper is a result of a short term collaboration between AB, AK and MS supported by the Programme DFG SPP 2392 Visual Communication (ViCom), Frankfurt am Main, Germany. AB and JS are supported by the Cluster Development Program Language Challenges” funded by the University of Cologne. JS is additionally funded by the DFG Collaborative Research Centre (CRC 1252) “Prominence in Language” based at the University of Cologne. AK is funded by DFG Priority Programme 2392 Visual Communication (ViCom, 2022-2025) grant awarded to AB. MS’ contribution to the publication has been produced in the context of the joint research funding of the German Federal Government and Federal States in the Academies’ Programme, with funding from the Federal Ministry of Education and Research and the Free and Hanseatic City of Hamburg. The Academies’ Programme is coordinated by the Union of the Academies of Sciences and Humanities. We thank Lina Herrmann for assistance with some annotations."

"German Science Foundation (DFG) SPP 2392 Visual Communication (ViCom)"

5. We note that your Data Availability Statement is currently as follows: "All relevant data are within the manuscript and its Supporting Information files."

6. We note that Figures 1-3 includes an image of a participant in the study. 

**Additional Editor Comments:**

Thanks for submitting this very interesting manuscript. Please revise the manuscript to address the reviewers' comments and provide documentation of your responses in a separate document, and I will render a decision without resending the manuscript to the reviewers.

Reviewers' comments:

Reviewer's Responses to Questions

**Comments to the Author**

1. Is the manuscript technically sound, and do the data support the conclusions?

Reviewer #1: Yes

Reviewer #2: Yes

2. Has the statistical analysis been performed appropriately and rigorously? 

Reviewer #1: Yes

Reviewer #2: Yes

3. Have the authors made all data underlying the findings in their manuscript fully available?

Reviewer #1: Yes

Reviewer #2: Yes

4. Is the manuscript presented in an intelligible fashion and written in standard English?

Reviewer #1: Yes

Reviewer #2: Yes

5. Review Comments to the Author

Reviewer #1: Review of Bauer et al., PONE-D-24-04524, PLOS One, Feburary 2024

Summary:

The present manuscript investigates head nods in natural data from the German Sign Language (DGS) corpus and aims to identify phonetic differences between nods serving distinct pragmatic functions. The authors use a combination of manual annotations and pose-estimation data to quantify potential phonetic differences between affirmative nods and those signaling feedback. They find that affirmative nods are produced faster and higher in amplitude as well as accompanied by manual lexical items. The authors conclude that the differences they identified are not peculiar to DGS but follow patterns that have previously been identified for ASL as well as English.

General assessment:

This is an interesting and methodologically innovative study of non-manual elements in DGS which is certainly worthy of publication after undergoing some revisions. The adopted approach is original and a nice combination of qualitative and quantitative methods. For the most part, all of the authors choices with regard to data selection and analyses are describe clearly. In addition, the authors make their data as well as analysis code available online for other researchers, thereby adhering to open science principles which are of special importance when working with data from a minority language such as DGS. While I have not re-run the analyses, I have had a look at the provided data and scripts and they are well-organized and documented.

The only thing I found a bit hard to follow while reading the manuscript was that the materials used in the study are described before the annotations are explained. So, I already knew before that there would be two types of head nods because this requirement led the authors to actually include additional data, but only in the section after the “materials” section there were details on how these annotations were created in the first place and how types of head nods were distinguished. To me it would make more sense to start with the criteria used for annotations, then describe the materials in the next section (including the information as to why the dataset had to be expanded in an unanticipated manner, i.e. due to the imbalance in observations), and then continue with the pose estimation section.

Focusing on the section about data selection, the general description as to how data was selected initially and how the dataset was then expanded is a bit sloppy and hard to follow. I understand that the authors took this step due to the uneven distribution of the types of head nods in the initial dataset, but this choice also makes the distribution of both types of head nods in the dataset meaningless (which should be clearly pointed out). I wonder if a pregistration of the desired annotations and minimum required data points for conducting meaningful statistical analyses would have helped the authors here to tell their story more elegantly, instead of more or less randomly selection video files and ending up with a very uneven distribution of head nod types and then adding just one type of nod data to be able to do statistics? Because the authors are interested in the phonetic differences between nods their approach is not a problem. But maybe preregistration of descried minimum sample sizes per type, etc. or similar is something to consider for future work?

Again relating to the annotations, a minor issue I noticed was that the authors provide inter-rater agreement for the length of annotations, but not for their coding of function and turn taking. I wondered why that was not done and/or why this information is not provided given that, for example, around 150 out of about 650 annotations of head not function were coded as “other”?

Lastly, another minor concern I had while reading was that sometimes I had the feeling that the manuscript currently lacks concision and, in parts, is written in a too narrative style with many fundamental pieces of information repeated again and again in different sections. I have provided some notes for parts that felt highly repetitive as part of my line-by-line comments below and hope that it will help the authors to make their manuscript more concise. However, please consider that I did not list all parts where I had the feeling that the text could be more concise or felt repetitive.

I have also listed other small and/or specific comments below with the aim to help the authors improve their manuscript before resubmission. Provided that the authors are willing to make their text more concise and incorporate my comments wherever they see fit, I recommend that such a revised version of the manuscript should be accepted for publication (in that case, I would not need to see such a revised version but leave that at the discretion of the editor).

Line-by-line comments:

Abstract: I’d spell out “German Sign Language” and give the abbreviation “DGS” in brackets, but I guess that’s a matter of taste and/or requirements of the journal’s stylesheet.

ln. 2: I am not a native speaker, but my intuition would be to put this into plural, so make it “head nods are …” and “interactions”? But if the authors are native speakers or have consulted with a native speaker, please disregard this comment.

ln. 220: As the abbreviation “DGS” has already been introduced above it should be used consistently.

ln. 220-232: Is all this background information on the corpus really relevant, given that you use only (parts) of the publicly available data anyway?

ln. 245-247: How were these samples drawn? Did the authors employ some kind of random sampling procedure or similar? If not, why not?

ln. 306: The author write “two tags were identified as separate movements if the offset of one tag and the 306 onset of the next were at least 300ms apart“. Does that imply that annotations occurring closer together in time were considered to be part of the “same” head nod?

ln. 391-399: I feel like I already know this from your extensive introduction and background, does it have to be repeated here?

ln. 408-415: I feel like this entire paragraph could be half a sentence as part of the previous one making it clear that these techniques are limited to 2D-data.

ln.420-427: This also reads like a repetition of information already given in the introduction. You already explained what you’re going to do in principle, so I think it would be fine to limit yourself to the actual methods and approach that you employed.

Figure 4: This is only a suggestion: Instead of reproducing these default images, it might be nicer to create a figure that shows the model and keypoints on top of a representative frame of your data. But please disregard if you consider this unnecessary or too much work.

ln. 489-494: I find this discussion a bit weird, what are these percentages supposed to tell the reader? You stated in the “materials” section that feedback nods were extremely overrepresented in your original sample, which is why you added additional data in which you only annotated affirmative nods. The relative percentages here then do not really provide any information to the reader other than how often these nods occurred in your selected subset of data (which doesn’t follow any objective selection criteria but was specifically expanded to kind of balance the occurrence of the two types of nods). In its current form this kind of reads to me as if these percentages would tell the reader something meaningful about the occurrence of these types of nodes “in the wild”, whereas that is not the case because a part of the dataset was not annotated for both types of head nod.

ln. 509: If I read the table correctly, what you are reporting here in the running text is the median not the mean?

ln. 529-536: This is very repetitive, as it has already been explained above at least once. Maybe cut?

Table 4: Are the t-test really meaningful here? That is, is the assumption of normality met? Given that you also ran non-parametric tests I assume that it’s not the case. So, what is the reason for reporting them, respectively reporting both?

ln. 618-632: What is the relationship between the co-occurrence (or lack thereof) of manual signs as well as mouthings with the signers turn-taking behavior? That is, it would be interesting to see what is the relationship between affirmative nods which mostly are not accompanied by manual lexical elements to turn taking—perhaps there is a relationship between presence of manual signs and initiating a new turn or similar? I don’t want to force the authors to explore this, but it might be food for thought.

ln. 647: “we find reveal …” –Please rephrase.

ln. 681-682: Yes, but is this really surprising given that you yourself already cite studies with hearing speaker of English that apparently show the same distinction in head nods (ln. 706)? Personally, I would not phrase all these findings as specific to DGS but rather discuss them from a more general perspective of communicative interactions which integrate audiovisual information in hearing speakers and layers different and partially simultaneously occurring visual information in deaf signers. That is, I don’t really see a reason to assume any differences here a priori, instead signed and spoken interactions just may simply constitute different use cases of the same pragmatic behaviours and signals.

ln. 707-744: This seems a bit too extensive, everything was already state in detail in the running text before. So, I would cut that a bit short really just pointing to all potential shortcomings. Similarly, this is a matter of style once again, but I would not end the manuscript with the section “Limitations”. Either add a short concluding paragraph after the limitations section or rearrange things a little bit in some other way.—After all you did good work, so that should also be the final message with which to leave the reader (imho).

Reviewer #2: It is an interesting research, which shows that head nods functioning as affirmative responses

differ in their phonetic properties and their alignment with manual elements from those head nods, which signal feedback in interaction. I have some minor comments:

a. Mouthing is mentioned by the authors (612-613 and in Footnote 5), but we do not see any future development of this case. How is it (MG) distributed between these two types fon head nods?

b. The feedback head nod (especially passive recipiency signals) must be considered as extralinguistic, as they are not grammatical participants of the signing text, or in the other words, they are not linguistically functioning elements on the morpho-syntactic level. I would recommend to add some linguistic discussion about this issue in order to outline the meaning of the presented research and to improve its theoretical frame.

c. There is a lack of wide typological picture, which could lead the authors to some theoretical discussion. Although the authors mentioned about it in subchapter ‘Limitations of this study and future work’, saying that crosslinguistic and cross-modal differences in terms of phonetic properties of head nods might be expected and they intend to address this in future research. (74)

6. PLOS authors have the option to publish the peer review history of their article (what does this mean?). If published, this will include your full peer review and any attached files.

Reviewer #1: No

Reviewer #2: **Yes: **Tamar Makharoblidze

---

## [Author Response · Author response to Decision Letter 0]

25 Apr 2024

A rebuttal letter 

that responds to each point raised by the academic editor and reviewer(s)

PONE-D-24-04524: “Phonetic differences between affirmative and feedback head nods in German Sign Language (DGS): A pose estimation study”

Journal Requirements:

I. Answers to Editor’s comments

Answer: We checked the document against PLOS ONE's style requirements and found the following inconsistencies:

a) we were referring to figures as “Fig. X”, this has been changed to the following format: “Fig X”,

b) the text was not double-spaced, which has been fixed.

Answer: The following “Funding Information” should be updated in the online submission system on our behalf please:

This work was supported as a short term collaboration grant between Anastasia Bauer, Anna Kuder and Marc Schulder supported by the Programme DFG SPP 2392 Visual Communication (ViCom), Frankfurt am Main, Germany. Anna Kuder is funded by a DFG grant awarded to Anastasia Bauer �project number 502013233� within the DFG Priority Programme 2392 Visual Communication (ViCom, 2022-2025).

Anastasia Bauer and Job Schepens are supported by the Cluster Development Program “Language Challenges” funded by the University of Cologne. Job Schepens is additionally funded by the DFG Collaborative Research Centre (CRC 1252) “Prominence in Language” �project number 281511265� based at the University of Cologne. 

Marc Schulder’s contribution to the publication has been produced in the context of the joint research funding of the German Federal Government and Federal States in the Academies’ Programme, with funding from the Federal Ministry of Education and Research and the Free and Hanseatic City of Hamburg. The Academies’ Programme is coordinated by the Union of the Academies of Sciences and Humanities. 

3. Please state what role the funders took in the study.

Answer: We include this amended Role of Funder statement in our cover letter and thank you for changing it in the online submission form on our behalf. Our Funding Statement should be please updated as follows:

This work was supported as a short term collaboration grant between Anastasia Bauer, Anna Kuder and Marc Schulder supported by the Programme DFG SPP 2392 Visual Communication (ViCom), Frankfurt am Main, Germany. Anna Kuder is funded by a DFG grant awarded to Anastasia Bauer �project number 502013233� within the DFG Priority Programme 2392 Visual Communication (ViCom, 2022-2025).

Anastasia Bauer and Job Schepens are supported by the Cluster Development Program “Language Challenges” funded by the University of Cologne. Job Schepens is additionally funded by the DFG Collaborative Research Centre (CRC 1252) “Prominence in Language” �project number 281511265� based at the University of Cologne. 

Marc Schulder’s contribution to the publication has been produced in the context of the joint research funding of the German Federal Government and Federal States in the Academies’ Programme, with funding from the Federal Ministry of Education and Research and the Free and Hanseatic City of Hamburg. The Academies’ Programme is coordinated by the Union of the Academies of Sciences and Humanities. 

Please remove any funding-related text from the manuscript and let us know how you would like to update your Funding Statement.

Answer: We have rewritten the “Acknowledgements Section” of our manuscript to: “We thank our student assistant, Lina Herrmann, for assistance with head nods annotations in one file."

5. Please confirm at this time whether or not your submission contains all raw data required to replicate the results of your study.

Answer: Our Data Availability Statement should be updated to read as follows: 

"The annotations and code produced in the context of this article are publicly available at

https:doi.org/10.5281/zenodo.10838847). The archive contains the following:

• the manual annotations created for the article in ELAN (.eaf) format;

• the Python code used for quantitative analysis of the annotations with the aid of OpenPose data from MY DGS – annotated; 

• the R code used to perform the statistical analyses discussed in the article as well as the depicted graphs

6. We note that Figures 1-3 includes an image of a participant in the study. 

Answer: All images in the article depicting persons are taken from the Public DGS Corpus, the dataset that this study is based on. Its license conditions permit the use of video and image materials for use in linguistic research studies and publications. These conditions are in line with the informed consent that the dataset creators received from participants. The consent forms are not publicly available, but the following documentation clarifies that our use of the data is covered: https://doi.org/10.25592/uhhfdm.1745

7. Please review your reference list to ensure that it is complete and correct. If you have cited papers that have been retracted, please include the rationale for doing so in the manuscript text, or remove these references and replace them with relevant current references. Any changes to the reference list should be mentioned in the rebuttal letter that accompanies your revised manuscript. If you need to cite a retracted article, indicate the article’s retracted status in the References list and alo include a citation and full reference for the retraction notice.

Answer: The following 4 entries at the reference list featured broken dois, which are now fixed: 

1. Napier J, Carmichael A, Wiltshire A. Look-Pause-Nod: A linguistic case study of a Deaf professional and interpreters working together. In: Hauser PC, Finch KL, Hauser AB, editors. Deaf professionals and designated interpreters: A new paradigm. Gallaudet University Press; 2008. p. 22–42.

2. Tanaka H. Turn-Taking in Japanese Conversation: A Study in Grammar and Interaction. John Benjamins Publishing; 2000

3. Östling R, Börstell C, Courtaux S. Visual Iconicity Across Sign Languages: Large-Scale Automated Video Analysis of Iconic Articulators and Locations. Frontiers in Psychology. 2018;9

4. Landis JR, Koch GG. The Measurement of Observer Agreement for Categorical Data. Biometrics. 1977;33(1):159–174.

To the following 4 entries we have added urls or dois, which were missing:

1. Wilbur RB. Non-manual markers: Theoretical and experimental perspectives. In: Quer J, Pfau R, Herrmann A, editors. The Routledge Handbook of Theoretical and Experimental Sign Language Research. London: Routledge; 2021. p. 530–565.

2. Schegloff EA. Discourse as an interactional achievement. Some uses of ’uh huh’ and other things that come between sentences. In: Tannen D, editor. Analyzing Discourse: Text and Talk. No. 19981 in Georgetown University Round Table on Languages and Linguistics. Washington, D.C.: Georgetown University Press; 1982. p. 71–93.

3. Gardner R. When Listeners Talk: Response tokens and listener stance. vol. 92 of Pragmatics & Beyond New Series. Amsterdam: John Benjamins Publishing Company; 2001.

4. Liddell SK. American Sign Language Syntax. No. 52 in Approaches to Semiotics. Berlin: De Gruyter Mouton; 1980.

Additionally, the current version of the paper includes three references which were not present in the previous version:

1. Clancy PM, Thompson SA, Suzuki R, Tao H. The conversational use of reactive tokens in English, Japanese, and Mandarin. Journal of Pragmatics. 1996;26(3):355–387

2. Jefferson G. Notes on a systematic deployment of the acknowledgement tokens “Yeah”; and “Mm Hm”. Paper in Linguistics. 1984;17(2):197–216.

3. Malisz, Zofia, and Maciej Karpiński. “Multimodal Aspects of Positive and Negative Responses in Polish Task-Oriented Dialogues.” Chicago, 2010.

In the rest of the refences we didn’t make any changes. None of the cited papers are retracted or in the retraction process.

II. Author’s notes

During the revision process we have noticed that our code and annotation files include some discrepancies when it comes to the number of occurrences of different types of nods. The changes were minimal, adding 1 or 2 values to some categories. We have eliminated those in the current submission version. This is why the current version features some changes to the values and measures reported previously. The values reported previously have been consistent with the code, but not with the ELAN annotation files. These changes are very slight and do not change our results or our narrative in any way. They however influence the fact that we need to apply changes to our figures (no. 5, 6, 7, 8, 9, 10, 11, 14) and tables (no. 4 and 5). The current version of the manuscript features the new tables and new figures and uploaded together with the current submission. The Zenodo link in the “supporting information” section now features the most current versions of the python and R codes as well as ELAN files. All of those resources yield numbers that are consistent with the values reported in the paper. 

We noticed that the red line which was mentioned in the description of Figure 3 is not visible on the actual illustration. The description was updated to correlate with the illustration.

III. Answers to the comments made by Reviewer #1:

General assessment by Reviewer #1:

Reviewer #1: The only thing I found a bit hard to follow while reading the manuscript was that the materials used in the study are described before the annotations are explained. So, I already knew before that there would be two types of head nods because this requirement led the authors to actually include additional data, but only in the section after the “materials” section there were details on how these annotations were created in the first place and how types of head nods were distinguished. To me it would make more sense to start with the criteria used for annotations, then describe the materials in the next section (including the information as to why the dataset had to be expanded in an unanticipated manner, i.e. due to the imbalance in observations), and then continue with the pose estimation section.

Answer: We have changed the order of those two sections. We start with the criteria used for annotations and then describe the DGS corpus data in the next section before we continue with the pose estimation. We have also made some minor adjustments to the text to better align with the new structure. This mostly covers removing a reference to data from "Annotation" and restructuring "Data" into having two subsections, "Source material" where we describe the corpus" and "Annotated material" where we describe what data we annotated and how/why we split it up.

Reviewer #1: Focusing on the section about data selection, the general description as to how data was selected initially and how the dataset was then expanded is a bit sloppy and hard to follow. I understand that the authors took this step due to the uneven distribution of the types of head nods in the initial dataset, but this choice also makes the distribution of both types of head nods in the dataset meaningless (which should be clearly pointed out). I wonder if a pregistration of the desired annotations and minimum required data points for conducting meaningful statistical analyses would have helped the authors here to tell their story more elegantly, instead of more or less randomly selection video files and ending up with a very uneven distribution of head nod types and then adding just one type of nod data to be able to do statistics? Because the authors are interested in the phonetic differences between nods their approach is not a problem. But maybe preregistration of descried minimum sample sizes per type, etc. or similar is something to consider for future work?

Answers:

We have revised the general overview about the initial data selection process and the subsequent expansion of the dataset to enhance clarity and coherence.

The point about the distribution of both types of head nods in the dataset being meaningless is a valid one. We realize that the fact that most of the feedback nods come from a different part of the dataset than the affirmative nods seems to be a limitation because we cannot make valid statements about distribution or frequency of different types of nods in the data and we point it out clearly. However, our main aim was not to talk about frequency of different types of head nods, as also highlighted by the reviewer 1, but about their phonetic properties, and for this we needed an augmented set of affirmation nods. There might have been a more elegant way of achieving this, we appreciate this being raised up and we will take it into consideration in future research by applying the method of preregistration. In the follow-up study, we apply a pregistration of the desired annotations and rather rely on the minimum required data points for conducting meaningful statistical analyses (which is 20 per cell according to the general recommendation), instead of a random selection video files and ending up with a very uneven distribution.

Reviewer #1: Again relating to the annotations, a minor issue I noticed was that the authors provide inter-rater agreement for the length of annotations, but not for their coding of function and turn taking.] I wondered why that was not done and/or why this information is not provided given that, for example, around 150 out of about 650 annotations of head not function were coded as “other”?

Answer: Thank you for this comment. It is true we are not providing IAR measures for coding for nods’ functions and turn-taking (TT) activity. That is because when it comes to coding for function and TT we had way more joint work before progressing to separate annotations and therefore our agreement must have been higher. Also, IAA measures wouldn’t work well here, as the annotations were not done blindly (that is, we discussed cases we were unsure about, which we haven't done for form coding before calculating IAA). What is more, the length of the function and TT annotation was exactly the same as for form (as they were produced on child tiers and the spans of the form annotations were copied exactly), so ELAN's modified kappa would yield a result that would be 'negatively' positive (too high), as it takes into consideration both overlap of annotations and their value. Lastly, on each of these tiers we only used two annotation values, which would also artificially strengthen our kappas. We have added this explanation to the manuscript. 

Reviewer #1: Lastly, another minor concern I had while reading was that sometimes I had the feeling that the manuscript currently lacks concision and, in parts, is written in a too narrative style with many fundamental pieces of information repeated again and again in different sections. I have provided some notes for parts that felt highly repetitive as part of my line-by-line comments below] and hope that it will help the authors to make their manuscript more concise. However, please consider that I did not list all parts where I had the feeling that the text could be more concise or felt repetitive.

Answer: Thank you for bringing this up. We have gone through the whole text of the manuscript and made it more concise by deleting the parts which were written in a too narrative style with many pieces being repeated in different sections. 

Reviewer #1: I have also listed other small and/or specific comments below with the aim to help the authors improve their manuscript before resubmission. Provided that the authors are willing to make 

---

## [Editor Report · Decision Letter 1]

6 May 2024

Phonetic differences between affirmative and feedback head nods in German Sign Language (DGS): A pose estimation study

PONE-D-24-04524R1

Dear Dr. Bauer,

We’re pleased to inform you that your manuscript has been judged scientifically suitable for publication and will be formally accepted for publication once it meets all outstanding technical requirements.

Kind regards,

Laura Morett

Academic Editor

PLOS ONE

Additional Editor Comments (optional):

Thank you for addressing the comments of the reviewers in your revision. I have reviewed your responses and revisions, and I conclude that this manuscript is now ready for publication. Therefore, I am pleased to accept it for publication at this time.
---

## [Editor Report · Acceptance letter]

15 May 2024

PONE-D-24-04524R1 

PLOS ONE

Dear Dr. Bauer, 

I'm pleased to inform you that your manuscript has been deemed suitable for publication in PLOS ONE. Congratulations! Your manuscript is now being handed over to our production team.

Kind regards, 

on behalf of

Dr. Laura Morett 

Academic Editor

PLOS ONE